# BiMark: Unbiased Multilayer Watermarking for Large Language Models

Xiaoyan Feng [* 1]   He Zhang [* 2]   Yanjun Zhang [3]   Leo Yu Zhang [1]   Shirui Pan [1]

## Abstract

Recent advances in Large Language Models (LLMs) have raised urgent concerns about LLM-generated text authenticity, prompting regulatory demands for reliable identification mechanisms. Although watermarking offers a promising solution, existing approaches struggle to simultaneously achieve three critical requirements: text quality preservation, model-agnostic detection, and message embedding capacity, which are crucial for practical implementation. To achieve these goals, the key challenge lies in balancing the trade-off between text quality preservation and message embedding capacity. To address this challenge, we propose BiMark, a novel watermarking framework that achieves these requirements through three key innovations: (1) a bit-flip unbiased reweighting mechanism enabling model-agnostic detection, (2) a multilayer architecture enhancing detectability without compromising generation quality, and (3) an information encoding approach supporting multi-bit watermarking. Through theoretical analysis and extensive experiments, we validate that, compared to state-of-the-art multi-bit watermarking methods, BiMark achieves up to 30% higher extraction rates for short texts while maintaining text quality indicated by lower perplexity, and performs comparably to non-watermarked text on downstream tasks such as summarization and translation.

## 1. Introduction

Large Language Models (LLMs) (OpenAI, 2022; Team, 2024b) have recently emerged in advancing cutting-edge technologies, and their rapid evolution has made LLM-generated content increasingly indistinguishable from human-created text. However, pressing security challenges (Zhang et al., 2023b; Wei et al., 2025; Gong et al., 2025; Luo et al., 2024; Zhang et al., 2025a;b; 2023a; 2024b; Pan et al., 2025) have been raised with the rapid progression of AI technologies (Zheng et al., 2024b; Zhang et al., 2024a; Zheng et al., 2024a; 2023; 2025; Guan et al., 2023), such as the misuse of LLMs (Tang et al., 2024) and synthetic data pollution (Shumailov et al., 2024). Therefore, recent legislation, such as the AI Act of the European Union (European Commission, 2023) and the executive order of the U.S. Department of Commerce (The White House, 2023), mandates the implementation of technical measures to mark and detect LLM-generated content. Thus, it appeals to LLM watermarking (Zhang et al., 2024e; Tang et al., 2024), a promising method to address these challenges, offering a proactive approach to embedding identifiable patterns in LLM-generated text (Liu et al., 2024).

Depending on whether it is integrated with the language model, watermarking can be categorized into inference-time watermarking and post-hoc watermarking (Tang et al., 2024). Compared to post-hoc methods that require an additional auxiliary module (Abdelnabi & Fritz, 2021; Yang et al., 2022; Yoo et al., 2023a), inference-time watermarking (Dathathri et al., 2024; Aaronson, 2023; Kirchenbauer et al., 2023) can embed detectable patterns linked to a secret key by modifying the LLM token sampling process in a plug-and-play manner, thereby efficiently achieving a cryptographically guaranteed detectable secret watermark (Aaronson, 2023; Kirchenbauer et al., 2023).

In general, implementing language model watermarking necessitates three key requirements: 1) *Text quality preservation*: The integration of watermarking mechanisms must maintain the fundamental utility and performance of language models (Hu et al., 2023; Kuditipudi et al., 2023). 2) *Model-agnostic detection*: The watermark verification process should function independently of the generating model's architecture, parameters, or access rights (Kirchenbauer et al., 2023; Kuditipudi et al., 2023). 3) *Message embedding capacity*: The watermarking scheme should accommodate sufficient capacity to encode crucial metadata such as model identity, generation timestamp, and content provenance (Bob & Dan, 2024; Pang et al., 2024).

Note that it is not trivial to satisfy the above three require-

[*]Equal contribution  [1]Griffith University, Brisbane [2]RMIT University, Melbourne [3]University of Technology Sydney, Sydney. Correspondence to: Leo Yu Zhang <leo.zhang@griffith.edu.au>, Shirui Pan <s.pan@griffith.edu.au>.

*Proceedings of the 42$^{nd}$ International Conference on Machine Learning*, Vancouver, Canada. PMLR 267, 2025. Copyright 2025 by the author(s).

*Table 1.* Comparison between BiMark (our method) and related studies.

| Techniques | | Methods | Requirements | | |
|---|---|---|---|---|---|
| Timing* | Vocabulary Partition | | Text Quality Preservation | Model-Agnostic Detection | Message Embedding Capacity |
| Post-Hoc | / | Yang et al. (2022), REMARK-LLM (Zhang et al., 2024c) | ● | ○ | ◐ |
| | | AWT (Abdelnabi & Fritz, 2021), Yoo et al. (2023a) | ● | ○ | ● |
| | | He et al. (2022a), CATER (He et al., 2022b) | ● | ● | ○ |
| Inference-Time | / | Aaronson (2023) | ◐ | ● | ○ |
| | | Kuditipudi et al. (2023), Zhao et al. (2024) | ● | ● | ○ |
| | | Fernandez et al. (2023) | ◐ | ● | ◐ |
| | Cumulative Probability | γ-reweight (Hu et al., 2023) | ● | ○ | ○ |
| | | DiPmark (Wu et al., 2023) | ● | ● | ○ |
| | Token Counting | GINSEW (Zhao et al., 2023) | ○ | ◐ | ○ |
| | | Soft Red List (Kirchenbauer et al., 2023) | ○ | ● | ○ |
| | | SynthID (Dathathri et al., 2024) | ● | ● | ○ |
| | | MPAC (Yoo et al., 2023b), Qu et al. (2024) | ○ | ● | ● |
| | | **BiMark (Ours)** | ● | ● | ● |

* Inference-time watermarking is achieved by modifying the sampling strategy of the model, such as adjusting the probability distribution of tokens. Post-hoc watermarking is achieved by editing existing text, such as synonym replacement and paraphrasing. In this table, ○ indicates "Not Considered", ◐ indicates "Partially Considered", and ● indicates "Considered". For multi-bit capacity, ◐ indicates methods require the prior knowledge of embedded messages, and ● indicates methods can work without such knowledge.

ments, as there is a trade-off between text quality preservation and message embedding capacity (Yoo et al., 2023b; Qu et al., 2024; Kirchenbauer et al., 2023). Some existing works (Yoo et al., 2023b; Qu et al., 2024) achieve efficient multi-bit watermarking by embedding one message bit per token through hashing. However, they rely on text quality-compromising zero-bit watermarking (Kirchenbauer et al., 2023) as a foundation, leading their enhancement of message embedding capacity inevitably come at the cost of text quality preservation. Note that, although other methods (Fernandez et al., 2023; Fairoze et al., 2023; Wang et al., 2023) can mitigate this issue to some extent by associating messages with secret keys or hash functions, they lack detection efficiency due to using messages non-invertibly.

To this end, we propose an integrated pipeline called BiMark to embed and extract multi-bit messages via watermarking while preserving text quality. BiMark requires neither training for embedding nor access to language models for detection. This approach simultaneously achieves the three key requirements. Our three-fold contributions are as follows:

1. For filling the gap between text quality preservation through unbiased reweighting and model-agnostic detection, we propose a token counting based reweighting approach that enables model-agnostic detection.

2. For handling the trade-off between message embedding capacity and text quality preservation, we present a multilayer reweighting mechanism that enhances detectability without sacrificing text quality.

3. For implementing multi-bit watermarking, we introduce an XOR-enhanced information encoding approach that enables messages carrying and message-agnostic extraction while preserving text quality.

## 2. Related Work

LLM watermarking aims to address the urgent need to differentiate model-generated text from human-written content. Text watermarking (Fang et al., 2017; Ziegler et al., 2019; He et al., 2022b; Fairoze et al., 2023; Fridrich et al., 2004; Christ et al., 2024; Zhao et al., 2024) has evolved significantly since its early application in content protection. Here we review related work on inference-time watermarking around the three key capacities in LLM watermarking.

**Text quality preservation**. Early watermarking methods such as Soft Red List (Kirchenbauer et al., 2023) significantly influenced the field but faced challenges with text quality preservation due to their biased nature. Recent work has addressed this challenge through two main approaches: unbiased reweighting (Hu et al., 2023; Wu et al., 2023) and distortion-free sampling (Kuditipudi et al., 2023; Zhao et al., 2024; Dathathri et al., 2024). Unbiased reweighting methods, such as γ-reweight (Hu et al., 2023) and DiPmark (Wu et al., 2023), modify the probability distributions of generated text to inject watermarks while ensuring that the expected probability distribution remains unchanged. Distortion-free sampling methods, such as Gumbel sampling (Aaronson, 2023; Kuditipudi et al., 2023) and SynthID (Dathathri et al., 2024), employ secret keys to guide the sampling process rather than modifying the probability distribution. Many of them enable model-agnostic detection, but lack multi-bit functionality.

**Model-agnostic detection**. The ability to detect watermarks without accessing the original or an auxiliary model is crucial for practical deployment. Existing watermarking methods developed from unbiased reweighting (Hu et al., 2023; Wu et al., 2023) utilize cumulative probability to

partition the vocabulary, which makes them unsuitable for model-agnostic detection. In the scenario of multi-bit watermarking, a similar concept critical for practical deployment is *message-agnostic detection*. That is, the ability to extract messages without access to the message space. Among existing multi-bit watermarking methods, only the works of Yoo et al. (2023b) and (Qu et al., 2024) satisfy this property through extracting messages from a voting matrix bit by bit. However, they are developed based on Soft Red List and thus without guaranteeing text quality.

**Message embedding capacity**. Message embedding capacity enables information to be embedded in and extracted from watermarked text, which is essential for tracing text provenance (Cohen et al., 2025; Pang et al., 2024). Recent studies develop multi-bit watermarking for message embedding capacity by associating each message with a unique secret key (Fernandez et al., 2023), or incorporating messages into hash functions as inputs (Fairoze et al., 2023; Wang et al., 2023). These methods require access to message space for extraction, as the messages themselves are entangled with hash functions in a non-invertible manner.

Yoo et al. (2023b) proposed a position allocation technique incorporating Soft Red List (Kirchenbauer et al., 2023), which can solve this problem. They allocate each token to a subunit of a message for watermark embedding, and extract the message bit by bit, which is in a message-agnostic way. Qu et al. (2024) further incorporates error correction codes to enhance multi-bit watermarking resilience. However, these methods were designed using Soft-List, leading to a lack of unbiasedness. As shown in the Tab. 1, a comprehensive comparison between BiMark and related studies is presented, highlighting our contributions in this field.

## 3. Preliminary

### 3.1. Watermarking for LLMs

**Text generation of LLMs**. A Large Language Model (LLM) produces text sequentially. Let $\mathcal{V}$ denote a vocabulary set of all tokens. A LLM generates a single token $x \in \mathcal{V}$ from a probability distribution $P_M \in \Delta_{\mathcal{V}}$ over all possible next tokens conditioned on preceding tokens, and continue this process autoregressively. Denoting a sequence of tokens $(x_1, x_2, \cdots, x_n)$ as $\boldsymbol{x}_{1:n}$, the joint generation probability of a token sequence can be written as: $P_M(\boldsymbol{x}_{n+1:n+m}|\boldsymbol{x}_{1:n}) = \prod_{i=1}^{m} P_M(x_{n+i}|\boldsymbol{x}_{1:n+i-1})$.

**Watermarking LLM-generated text**. To reliably distinguish between LLM-generated and human-written text, watermarking methods actively inject detectable signals using secret keys (Kirchenbauer et al., 2023; Aaronson, 2023) into LLM-generated text. During detection, through measuring the similarity between extracted signals and injected signals with the same keys, the watermark can be verified.

Early works (Kirchenbauer et al., 2023; Kuditipudi et al., 2023; Hu et al., 2023) only focus on addressing the binary question of whether a text contains watermarks, referred to *zero-bit watermarking*. For more reliable text authenticity, watermarking methods that can carry informative messages are proposed, referred to as *multi-bit watermarking* (Yoo et al., 2023b; Fairoze et al., 2023; Cohen et al., 2025).

A *zero-bit watermarking scheme* has two key components: 1) a distribution reweighting function $R_k : \Delta_{\mathcal{V}} \to \Delta_{\mathcal{V}}$ through which the key $k$ modifies the original probability distribution to inject signals, where $\Delta_{\mathcal{V}}$ is the set of all possible probability distributions over the vocabulary set $\mathcal{V}$; and 2) a detector: $D : \mathcal{V}^* \to \{\text{True}, \text{False}\}$ which determines whether a token sequence contains the watermarks, where $\mathcal{V}^*$ denotes all possible token sequences over vocabulary $\mathcal{V}$.

A *multi-bit watermarking scheme* has two components: 1) a distribution reweighting function $R_{k,\boldsymbol{m}} : \Delta_{\mathcal{V}} \to \Delta_{\mathcal{V}}$ that works similarly to the zero-bit scheme but additionally incorporates a message $m \in \{0, 1\}^{\ell}$ for watermarking, where $\ell$ is the message length. 2) a detector: $D : \mathcal{V}^* \to \{0, 1\}^{\ell}$, which is also similar to a zero-bit watermark detector but can extract embedded messages from watermarked text.

### 3.2. Soft Red List Watermarking

Soft Red List (Kirchenbauer et al., 2023) is a pioneering inference-time watermarking method, introducing the ingenious concept of *green list* and *red list* for LLM watermarking. We examine this method because its *green list* and *red list* approach inspired our bit-flip unbiased reweighting technique, which determines green lists through a coin flip.

**Green-Red partition for watermarking**. Soft Red List embedding watermarks by employing a uniquely devised vocabulary bipartition framework guided by a pseudorandom function $\text{prf}_k$. This function, initialized with a secret key $k$, takes a sliding window of previous tokens as input and generates a binary mask for vocabulary partitioning. The context-localized and key-dependent nature of $\text{prf}_k$ ensures both efficiency and tamper resistance of the watermark. Given a proportion $\gamma$, the method splits vocabulary $\mathcal{V}$ into a green list and a red list using $\text{prf}_k$ such that the size of green lists is $\gamma|\mathcal{V}|$. When generating text, the method constructs a probability distribution $P_{M,w}$ to facilitate watermarking by adjusting $P_M$ — enhancing probabilities for tokens on green lists while decreasing those on red lists.

**Z-test for watermark detection**. Soft Red List watermarks can be detected by comparing the proportion of green tokens in the text with the expected proportion $\gamma$, because the watermarked LLM is influenced to generate more tokens from green lists created by a secret key. Specifically, given a text segment with $T$ tokens, let $G$ denote the count of tokens on green lists. Since green list membership is

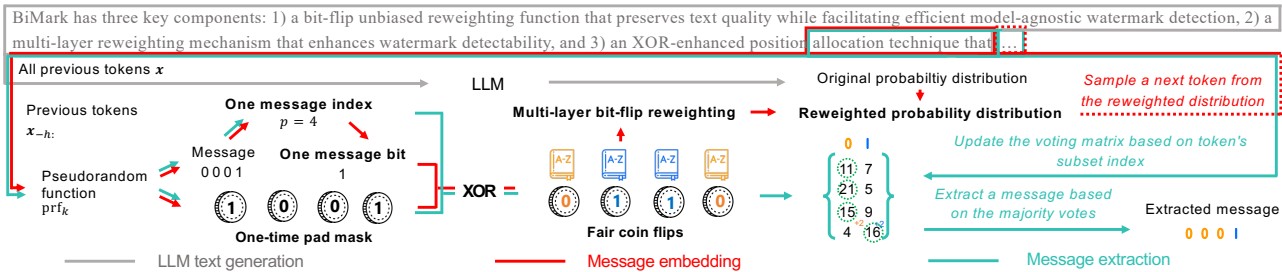

*Figure 1.* Pipeline of BiMark. a) The LLm outputs a probability distribution over all tokens (grey lines). b) The message embedding process (red lines) modifies this distribution using a pseudorandom function that selects message bits and generates one-time pad masks. After XOR operations create fair coin flips, multilayer unbiased reweighting guides token sampling. c) Message extraction (green lines) reconstructed the message by observing token subset memberships across multilayers and using majority voting to recover message bits.

randomly determined for each token, a non-watermarked text will have a green token proportion close to $\gamma$ following the law of large numbers. To statistically test for deviations, *a one-proportion z-test* can be applied:

$$z = \frac{G/T - \gamma}{\sqrt{\gamma(1-\gamma)/T}}.$$

If the z-score surpasses a predefined threshold, it suggests that the text is watermarked, since the proportion of green tokens deviates significantly from the expected value $\gamma$. When using the z-score for watermark detection, the corresponding p-value reflects the false positive rate (Type-I error), whereas the false negative rate (Type-II error) is influenced by the inherent entropy characteristics of the LLM (Kirchenbauer et al., 2023; Kuditipudi et al., 2023).

### 3.3. Watermarking via Unbiased Reweighting

While Soft Red List effectively embeds detectable watermarks, it potentially degrades text quality by introducing biases in the generation process. To address this limitation, Hu et al. (2023) proposed unbiased watermarking, which aims to maintain the original distribution of generated text while enabling watermark detectability. Assume that a service provider creates watermarks using a secret key $k$ randomly chosen from a key space $\mathcal{K}$ following a prior distribution $P_k(k)$. A desirable watermarking property is that the probability distributions of watermarked text and non-watermarked text are identical. This can be formally defined as follows:

**Definition 3.1** (*n*-shot undetectable (Hu et al., 2023)). For a fixed input sequence $\boldsymbol{a} \in \mathcal{V}^*$, we say that watermarked LLM and key prior pair $(P_{M,w}, P_k)$ is *n*-shot-undetectable compared to original LLM $P_M$ if for any $n$ number of strings $\boldsymbol{x}^i \in \mathcal{V}^*$:

$$\prod_{i=1}^{n} P_M(\boldsymbol{x}^i|\boldsymbol{a}) = \sum_{k \in \mathcal{K}} P_k(k) \prod_{i=1}^{n} P_{M,w}(\boldsymbol{x}^i|\boldsymbol{a};k).$$

To achieve 1-shot undetectability, Hu et al. (2023) proposed

*unbiased reweighting function* for a single token generation:

**Definition 3.2.** (Unbiased reweighting function (Hu et al., 2023)). Given a random variable $e$ and a reweighting function $R_e : \Delta_{\mathcal{V}} \to \Delta_{\mathcal{V}}$, we say that $R_e$ is an *unbiased* reweighting function if and only if for all probability distributions $P \in \Delta_{\mathcal{V}}$, $E_e[R_e(P)] = P$.

To achieve $n$-shot undetectability, they introduced a context tracking mechanism that only generates watermarked tokens when encountering new context tokens and generates non-watermarked tokens otherwise. This mechanism prevents the reuse of previously consumed context tokens as seeds for watermarking, thereby ensuring the independence of random variables used in the reweighting function.

### 3.4. Multi-bit Watermarking via Position Allocation

Many prior works (Fernandez et al., 2023; Fairoze et al., 2023; Wang et al., 2023; Yoo et al., 2023b; Qu et al., 2024) have explored multi-bit watermarking that extends beyond zero-bit detection. Among them, Yoo et al. (2023b) proposed Multi-bit Watermarking via Position Allocation (MPAC), which enables message embedding without additional latency compared to Soft Red List and allows message extraction in a message-agnostic manner. We examine this technique, as it inspired BiMark, which integrates it with XOR operations to achieve unbiased watermarking.

**Message embedding**. MPAC inherits the framework of Soft Red List and encodes messages into watermarks by selecting different vocabulary subsets as green lists. Assuming the message is a binary string $\boldsymbol{m} \in \{0,1\}^\ell$, for each token generation: A pseudorandom function $\text{prf}_k$ takes previous tokens in a sliding window to 1) partition the vocabulary $\mathcal{V}$ into two balanced subsets [1] $\mathcal{V}_0$ and $\mathcal{V}_1$, 2) select one position $p \in \{1, 2, \cdots, \ell\}$ which determines which message bit will be encoded. The method then designates the vocabulary subset $\mathcal{V}_{\boldsymbol{m}[p]}$ as the green list, and the subsequent process

---

[1] In the original paper, the number of partitions can be over 2. For ease of discussion, we take bipartition here.

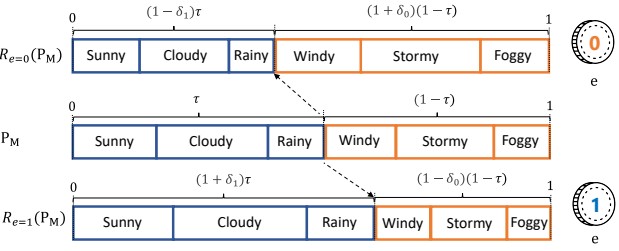

*Figure 2.* Bit-flip unbiased reweighting. Given a vocabulary bipartition $\mathcal{V}_0$ (shown in yellow) and $\mathcal{V}_1$ (shown in blue), a probability distribution $P_M$ over the vocabulary $\mathcal{V}$ is adjusted based on a fair coin flip $e$. When $e = 0$, probabilities of $\mathcal{V}_0$ is increased by $\delta_0\%$ and probabilities of $\mathcal{V}_1$ is decreased by $\delta_1\%$. When $e = 1$, probabilities of $\mathcal{V}_1$ is increased by $\delta_1\%$ and probabilities of $\mathcal{V}_0$ is decreased by $\delta_0\%$. This symmetric adjustment preserves original probabilities in expectation when the coin flip are fair.

of increasing probabilities following as Soft Red List.

**Message extraction**. MPAC extracts embedded messages from text using a green-list voting matrix $M \in \mathbb{R}^{\ell \times 2}$ in a message-agnostic manner. Given a text, for each token $x$, the method pseudorandomly reconstructs the corresponding vocabulary partition $\mathcal{V}_1$ and $\mathcal{V}_0$, along with the message position $p$. Next, it checks the subset membership of token $x$: if $x \in \mathcal{V}_0$, then $M[p][0]$ is incremented by 1; if $x \in \mathcal{V}_1$, then $M[p][1]$ is incremented by 1. After processing all tokens, based on the assumption that watermarked text contains more green tokens than red tokens, each bit of the extracted message can be obtained as $\arg\max(M[p][:])$ for $p \in \{1, \cdots, \ell\}$. The sum of the majority votes $\sum_{p=1}^{\ell} \max(M[p][:])$ serves as the total green tokens for determining whether the text contains watermarks.

## 4. The Proposed BiMark

BiMark is an unbiased multi-bit watermarking framework designed to achieve text quality preservation, message embedding capacity, and model-agnostic and message-agnostic detection simultaneously. It operates in two phases: watermark embedding and watermark detection. In the embedding phase, we introduce a novel bit-flip reweighting function that preserves text quality while facilitating model-agnostic detection. To enhance watermark detectability, a multilayer reweighting mechanism is employed. For carrying messages, we propose multi-bit watermarking via XOR-enhanced position allocation based on a one-time pad mechanism. In the detection phase, a voting matrix is constructed based on the text for message extraction. Fig. 1 illustrates the overall architecture of BiMark.

### 4.1. Bit-Flip Unbiased Reweighting

An unbiased reweighting function is the core component of unbiased watermarking. While unbiased reweighting helps

preserve text quality in watermarking, existing methods (Hu et al., 2023; Wu et al., 2023) based on cumulative probability distributions require access to model probabilities, making them unsuitable for model-agnostic detection.

We propose a bit-flip reweighting function that relies on token counting-based bipartitions, enabling both unbiased reweighting and model-agnostic detection. The core mechanism of which is straightforward: we use a *fair* coin flip to determine the direction of probability redistribution between two vocabulary bipartitions $\mathcal{V}_0$ and $\mathcal{V}_1$. When the coin shows heads, we increase the probabilities of tokens in $\mathcal{V}_1$ while proportionally decreasing the probabilities in $\mathcal{V}_0$, and vice versa for tails. This symmetric adjustment and the fairness of the coin flip ensure the expected token distribution remains unchanged while creating a detectable pattern in text generated by watermarked LLMs.

Formally, let a random variable $e$ represent a fair coin flip, where $e = 1$ indicates heads and $e = 0$ indicates tails. Given a vocabulary bipartition $(\mathcal{V}_0, \mathcal{V}_1)$, the bit-flip reweighting function Fig. 2 adjusts an original probability as follows:

$$R_{\theta,e}(P_M)(x) = \begin{cases} (1 + \delta_1)P_M(x) & \text{if } e = 1 \wedge x \in \mathcal{V}_1, \\ (1 - \delta_0)P_M(x) & \text{if } e = 1 \wedge x \in \mathcal{V}_0, \\ (1 - \delta_1)P_M(x) & \text{if } e = 0 \wedge x \in \mathcal{V}_1, \\ (1 + \delta_0)P_M(x) & \text{if } e = 0 \wedge x \in \mathcal{V}_0. \end{cases}$$
(1)

Here, $\delta_0$ and $\delta_1$ are scaling factors for $\mathcal{V}_0$ and $\mathcal{V}_1$, respectively, and $\theta = (\delta_0, \delta_1, \mathcal{V}_0, \mathcal{V}_1)$ denotes reweighting function configuration. Fig. 2 illustrates an example of the bit-flip unbiased reweighting. The scaling factors $\delta_0$ and $\delta_1$ must maintain a valid probability distribution. Subject to the constraint that the probability distribution sums to unity, a valid $\delta_0$ can be derived by a valid $\delta_1$ as Lemma 4.1.

**Lemma 4.1** (Scaling factor constraint). *Let $\tau = \sum_{x \in \mathcal{V}_1} P_M(x)$ be the total probability of partition $\mathcal{V}_1$ in the distribution $P_M$. For $\tau < 1$, the scaling factor of $\mathcal{V}_0$ must satisfy:*

$$\delta_0 = \delta_1 \cdot \tau / (1 - \tau). \tag{2}$$

*Proof.* Let $\Delta$ be the absolute change in probability between partitions $\mathcal{V}_0$ and $\mathcal{V}_1$. When $\Delta > 0$, for the probability distribution to remain valid:

$$\Delta = \delta_1 \tau = \delta_0 (1 - \tau),$$

It is clear to see that $\delta_0 = \delta_1 \cdot \tau / (1 - \tau)$. $\qquad \square$

Due to the constraint in Eq. (2), when $\delta_1$ ensures a valid probability distribution, $\delta_0$ also satisfies the corresponding bound. Although the unbiased reweighting function is established, implementing watermarking faces a challenge of determining appropriate scaling factors $\delta_0, \delta_1$ that maintain

a valid probability distribution while achieving effective watermarking. This requires carefully handling edge cases and constraints that arise in real-world scenarios.

For practical implementation, we introduce a base scaling factor $\tilde{\delta} \in [0, 1]$ and derive $\delta_1$ adaptively based on $\tau$ from Eq. (3). However, two scenarios[2] require special attention: (1) If $\tau = 0$, we set $\delta_1 = 0$ since all tokens in $\mathcal{V}_1$ have zero probability and any non-zero $\delta_1$ would have no effect on the probability distribution and will break the symmetry of our reweighting scheme when $e = 1$. (2) If $(1+\tilde{\delta})\tau > 1$, we set $\delta_1 = (1-\tau)/\tau$ which makes the probability of $\mathcal{V}_1$ become 1 and the probability of $\mathcal{V}_0$ become 0, ensuring the reweighted distribution remains a valid probability distribution over two partitions. Consequently, the scaling factor $\delta_1$ becomes:

$$\delta_1 = \begin{cases} 0 & \text{if } \tau = 0, \\ (1-\tau)/\tau & \text{if } (1+\tilde{\delta})\tau > 1, \\ \tilde{\delta} & \text{otherwise.} \end{cases} \quad (3)$$

It is easy to check this piecewise definition of $\delta_1$ ensures a valid probability distribution for all possible cases.

**Theorem 4.2** (Bit-Flip unbiased reweighting). *For any probability distribution over a vocabulary $\mathcal{V}$ and a reweighting function $R_{\theta,e}$ defined above, we have $E_e[R_{\theta,e}(P)] = P$.*

This property follows naturally from our design: we use fair coin flips to determine which partition's probabilities to increase or decrease, and ensure the probability changes are equal in magnitude. The symmetry of these adjustments guarantees unbiasedness while enabling watermark detection through token count analysis—a key advantage over previous cumulative probability-based approaches. The proof of Theorem 4.2 can be found in App. A.1. Note that both $\mathcal{V}_0$ and $\mathcal{V}_1$ have equal probabilities of being the green list due to fair coin flips. To obtain a consistent proportion for the z-test, both sets should have equal size: $|\mathcal{V}_0| = |\mathcal{V}_1| = |\mathcal{V}|/2$.

### 4.2. Multilayer Unbiased Reweighting

In Soft Red List (Kirchenbauer et al., 2023), watermark detectability can be enhanced by combining a smaller green list proportion $\gamma$ with a larger logit adjustment value $\delta$. Since Soft Red List imposes no constraints on the unbiased property, $\delta$ can be arbitrarily large—in the extreme case, restricting generation to only green list tokens (Kirchenbauer et al., 2023). In contrast, the bit-flip reweighting is constrained by unbiased probability adjustment—with only a single layer reweighting (one bipartition), the watermark may be weak and difficult to detect due to an imbalanced probability distribution. To address this limitation, we propose a multilayer reweighting mechanism that preserves the unbiased property while enhancing watermark detectability through multiple

---

[2]For ease of presentation, we assume $(1+\delta_1)\tau > (1+\delta_0)(1-\tau)$ and focus on the edge outlier caused by expanding $\mathcal{V}_1$.

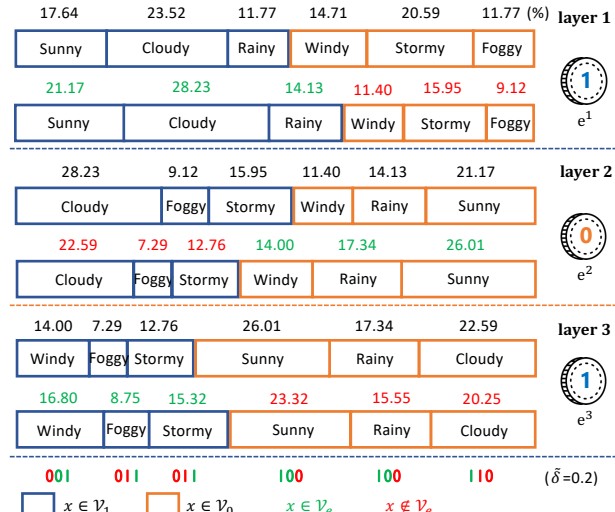

*Figure 3.* Multilayer bit-flip reweighting. Given multiple bipartitions, each reweighting layer adjusts the probability from the previous reweighting layer based on the bipartition and the fair coin flip result of this layer. In layer 1, "Sunny" belongs to $\mathcal{V}_1$ and $e = 1$, making "Sunny" become green and gain probability. In layer 2, "Sunny" belongs to $\mathcal{V}_0$ and $e = 0$, making "Sunny" become green and gain probability. In layer 3, "Sunny" belongs to $\mathcal{V}_1$ and $e = 1$, making "Sunny" become red and lose probability.

independent bipartitions and fair coin flips to iteratively adjust probability distributions, as shown in Fig. 3.

Formally, given a sequence of independent vocabulary bipartitions $[(\mathcal{V}_0^1, \mathcal{V}_1^1), \cdots, (\mathcal{V}_0^d, \mathcal{V}_1^d)]$ and a sequence of independent fair coin flips $\boldsymbol{e} = [e^1, e^2, \cdots, e^d]$, i.e. $e^i$ follows Bernoulli(0.5), the multilayer reweighting is defined as:

$$P_{M,w} = R_{\theta^d,e^d} \circ R_{\theta^{d-1},e^{d-1}} \circ ... \circ R_{\theta^1,e^1}(P_M),$$

where $\circ$ denotes function composition (applied from right to left), and $R_{\theta^i,e^i}$ is a bit-flip reweighting function configured with the $i$-th bipartition, the $i$-th fair coin flip, and a base scaling factor $\tilde{\delta}$ used to obtain a valid $\delta_0$ and a valid $\delta_1$ for unbiased reweighting as described in Section 4.1.

When vocabulary bipartitions $(\mathcal{V}_0^i, \mathcal{V}_1^i)$ and fair coin flips $e_i$ are independent across layers, the multilayer reweighting is still unbiased, i.e., $E_{\boldsymbol{e}}[P_{M,w}] = P_M$. A discussion and proof of this property can be found in App. A.3.

### 4.3. XOR-enhanced Position Allocation

Integrating with the multilayer unbiased reweighting mechanism, we propose an unbiased multi-bit watermarking method via XOR-enhanced position allocation technique, which can embed messages into text while maintaining text quality and can extract the message from the text.

**Message embedding**. To embed a message $\boldsymbol{m} \in \{0, 1\}^\ell$ into generated text, we pseudorandomly encode it into a

sequence of balanced bits as fair coin flips $e$ that guides our multilayer reweighting. This encoding process is achieved through a one-time pad with XOR operations. For a single token generation, fair coin flips $e$ are constructed as follows:

1) *Message bit selection:* A message index $p \in \{1, \cdots, \ell\}$ is pseudorandomly selected using previous tokens in a sliding window as a seed. Then, $\boldsymbol{m}[p]$ is one bit of the message that will be embedded in the generated token.

2) *Random mask generation:* A sequence of balanced bits $\boldsymbol{b} = [b^1, b^2, \cdots, b^d]$ is pseudorandomly sampled from Bernoulli$(0.5)$ using previous tokens in a sliding window as a seed. These bits serve as a one-time pad mask.

3) *Fair coin flip computation:* To encode the message bit, for each layer, a fair coin flip is calculated by $e^i = \boldsymbol{m}[p] \oplus b^i$, where $\oplus$ is the XOR operation and $i \in \{1, 2, \cdots, d\}$. These fair coin flips are used in the multilayer bit-flip reweighting.

This process ensures two crucial properties. First, the message is recoverable because the XOR operation is reversible — given $b^i$ and $e^i$ for each token, the corresponding message bit $\boldsymbol{m}[p]$ can be recovered. Second, and equally important, the unbiased property is guaranteed. This is because XORing any fixed bit (in our case, $\boldsymbol{m}[p]$) with a random bit sampled from Bernoulli$(0.5)$ produces a result bit that also follows Bernoulli$(0.5)$ (see App. A.4 for proof) — a property that ensures fair coin flips for unbiased reweighting regardless of the message content. See App. C for the complete watermarked text generation algorithm.

Note that in this encoding process, message bits and one-time pad masks are contructed pseudorandomly for each token during text generation. The vocabulary partitions used in unbiased reweighting are constructed pseudorandomly before the entire text generation process. Compared to existing methods which construct vocabulary partitions or permutation for each token (Kirchenbauer et al., 2023; Hu et al., 2023; Wu et al., 2023), this approach is more efficient.

**Message extraction**. The embedded message $\boldsymbol{m}$ is extracted from the watermarked text by analyzing the statistical patterns created by the multilayer reweighting process. The approach is extended from the voting matrix method introduced in Section 3.4 adapting the multilayer reweighting and the XOR operations. Given a text segment, votes of message bits are gathered from each token $x$ as follows:

1) *Recovering variables used by encoding:* The message index $p$ and the one-time pad mask $\boldsymbol{b} = [b^1, b^2, \cdots, b^d]$ used in the generation phase are reconstructed pseudorandomly using the same key. Note the message $\boldsymbol{m}[p]$ is unknown but can be reconstructed if the $e^i, i \in \{1, 2, \cdots, d\}$ is known.

2) *Gathering votes of message bits:* To reconstruct the message bit, for each layer $i$, the fair coin flip $e^i$ is estimated through tokens' subset membership as follows:

$$\hat{e}^i = \begin{cases} 0 & \text{if } x \in \mathcal{V}_0^i, \\ 1 & \text{if } x \in \mathcal{V}_1^i. \end{cases}$$

This estimation leverages the statistical bias introduced during generation — tokens were more likely from partitions corresponding to the respective coin flip results.

The message bit $\boldsymbol{m}[p]$ is estimated by $\hat{e}^i \oplus b^i$, which is a reversal of the XOR operation, and is taken as a vote of the message bit. This vote is accumulated by incrementing $\boldsymbol{M}[p][\hat{e}^i \oplus b^i]$ by 1. The multilayer design provides $d$ independent votes per token, enhancing extraction reliability.

3) *Extracting message bits through majority voting:* After all tokens are processed, the message is extracted through the majority vote bit by bit as follows:

$$\boldsymbol{m}[p] = \arg\max(\boldsymbol{M}[p][:]) \text{ for } p \in \{1, 2, \cdots, \ell\}.$$

The extraction process reliably recovers the embedded message by leveraging both the deterministic nature of the pseudorandom function and the statistical patterns created by our multilayer reweighting scheme. See App. C for the complete watermark detection algorithm.

In summary, BiMark achieves unbiased multi-bit watermarking through a carefully orchestrated pipeline. During generation, the multilayer unbiased reweighting mechanism applies iterative probability adjustments guided by fair coin flips where message bits are encoded, which creates detectable statistical patterns while preserving the original generation distribution. During detection, these patterns are verified by analyzing token distribution across bipartitions of multiple layers and aggregating evidence in a voting matrix, enabling message extraction. The framework maintains text quality while enabling message embedding capacity with model-agnostic and message-agnostic detection.

## 5. Experimental Analyses

We conduct comprehensive experiments to evaluate BiMark's effectiveness across three key dimensions: message embedding capacity, text quality preservation, and an ablation study of the multilayer mechanism. For comparisons, we focus on inference-time methods that are publicly available and support model-agnostic and message-agnostic detection. The code is available at: https://github.com/Kx-Feng/BiMark.git.

### 5.1. Message Embedding Capacity

**Experimental setup**. For experiments of message embedding capacity, the Llama3-8B model (AI@Meta, 2024) is used for text generation with temperature 1.0 and top-50 sampling. C4-RealNewslike (Raffel et al., 2020) dataset is used as prompts. For pseudorandom operations of watermarking methods, a 2-token sliding context window is

*Table 2.* Message extraction rate.

| Bits | Method | Text Length (Tokens) | | | | | | | |
|---|---|---|---|---|---|---|---|---|---|
| | | 50 | | 100 | | 200 | | 300 | |
| | | Rate | PPL | Rate | PPL | Rate | PPL | Rate | PPL |
| 8 | BiMark | **95.26** | **8.5** | **97.62** | 7.38 | **98.15** | 5.78 | 97.88 | **4.57** |
| | MPAC (1) | 49.49 | 9.44 | 79.72 | 8.64 | 89.51 | 8.08 | 93.48 | 7.85 |
| | MPAC (1.5) | 78.81 | 9.76 | 89.75 | 9.13 | 96.4 | 8.63 | **98.57** | 8.39 |
| 16 | BiMark | **85.55** | **8.60** | **93.31** | 7.35 | **95.54** | 5.83 | **95.54** | **4.71** |
| | MPAC (1) | 57.04 | 9.36 | 68.25 | 8.63 | 79.31 | 8.17 | 87.50 | 7.89 |
| | MPAC (1.5) | 66.06 | 9.69 | 78.21 | 8.87 | 89.04 | 8.45 | 93.83 | 8.45 |
| 32 | BiMark | **66.35** | **8.63** | **82.69** | 7.48 | **89.68** | 5.89 | **90.22** | **4.75** |
| | MPAC (1) | 45.06 | 9.16 | 56.79 | 8.34 | 67.96 | 7.92 | 74.32 | 7.67 |
| | MPAC (1.5) | 51.03 | 9.77 | 65.35 | 8.96 | 78.15 | 8.49 | 84.70 | 8.35 |

[*] "Rate"(↑) denotes message extraction rate which is the ratio of correctly extracted bits, and "PPL" (↓) denotes perplexity. "MPAC (1)" and "MPAC (1.5)" indicate that the value added to green tokens' logit scores are $\delta = 1$ and $\delta = 1.5$, respectively.

*Table 3.* 8-Bit message extraction rate from damaged text.

| Ratio | Method | Length | | | | | | |
|---|---|---|---|---|---|---|---|---|
| | | 25 | 50 | 100 | 150 | 200 | 250 | 300 |
| 0.1 | BiMark | **70.25** | **82.4** | **91.02** | **93.74** | **94.94** | **95.23** | **95.24** |
| | MPAC (1) | 50.38 | 59.5 | 67.08 | 73.41 | 76.34 | 80.15 | 82.1 |
| | MPAC (1.5) | 56.42 | 66.8 | 76.85 | 82.64 | 86.55 | 89.33 | 91.62 |
| 0.2 | BiMark | **60.61** | **71.6** | **81.71** | **86.29** | **88.62** | **89.39** | **90.01** |
| | MPAC (1) | 45.51 | 52.83 | 60.72 | 64.29 | 67.31 | 70.03 | 72.19 |
| | MPAC (1.5) | 50.04 | 58.04 | 67.36 | 73.1 | 76.37 | 78.99 | 80.85 |
| 0.3 | BiMark | **53.37** | **62.52** | **70.37** | **74.22** | **76.23** | **77.3** | **78.23** |
| | MPAC (1) | 43.18 | 48.71 | 53.36 | 56.3 | 58.29 | 59.78 | 60.57 |
| | MPAC (1.5) | 45.59 | 51.16 | 58.11 | 62.24 | 64.73 | 67.07 | 68.31 |

used for seeding. For evaluating the generation quality of language models with watermarks, perplexity of generated text is calculated using Gemma-9B (Team, 2024a) as an oracle model which has more parameters than Llama3-8B. The baseline methods include MPAC (Yoo et al., 2023b), Soft Red List (Kirchenbauer et al., 2023), and SynthID (Dathathri et al., 2024). For MPAC and Soft Red List, the proportion $\gamma$ of green lists is 0.5 for a balance between detectability and text quality, and also fair comparison with BiMark's vocabulary bipartiiton setting. For SynthID, the number of tournaments is 30, as recommended in the default setting. For BiMark, the base scaling factor $\tilde{\delta}$ is 1.0, and the number of layers $d$ is 10, which provides a balance between detectability and computational efficiency.

**Multi-bit watermarking scenario.**

In this experiment, we compare BiMark with MPAC (Yoo et al., 2023b), a state-of-the-art multi-bit watermarking method that is model-agnostic and message-agnostic detectable. We test with varying message lengths (8, 16, and 32 bits) and text lengths (50, 100, 200, and 300 tokens), using message extraction rate, i.e. the bit accuracy between embedded messages and extracted messages, as the evaluation metric. As shown in Tab. 2, MPAC significantly sacrifices text quality when the value $\delta$ added to logit scores of green tokens grows from 1 to 1.5, even though it improves the message extraction. In contrast, BiMark consistently achieves higher extraction rates with lower perplexity. For 50-token texts, BiMark improves message extraction rates by 20.87%, 29.50%, and 30.02% for 8-bit, 16-bit, and 32-bit messages respectively, compared to MPAC. The per-

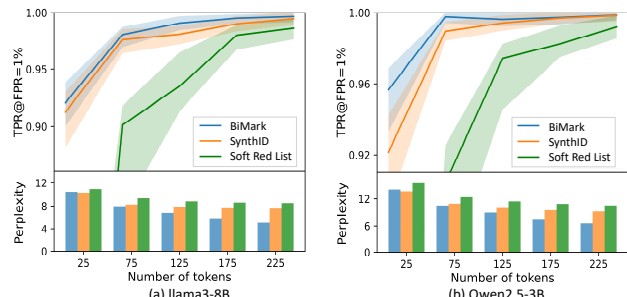

*Figure 4.* Zero-bit Watermark Detection.

formance improvement is more evident with both shorter texts and longer messages, which typically present greater challenges in watermarking.

We further evaluate resilience against synonym substitution attacks (Jovanović et al., 2023; Zhang et al., 2024d; Hou et al., 2023; Ren et al., 2023) using BERT (Devlin et al., 2018) and WordNet (Miller, 1995). Tab. 3 demonstrates BiMark's superior performance—for 100-token texts, BiMark maintains 18.44%, 21.30%, and 26.24% higher extraction rates than those of MPAC under 0.1, 0.2, and 0.3 text substitution ratios, respectively. This enhanced resilience is attributed to the fact that the multilayer mechanism provides abundant watermark evidences, leading the statistical patterns of detection to be more distinguishable between watermarked and non-watermarked text.

**Zero-bit watermarking scenario.** We access BiMark's performance on zero-bit watermarking by embedding 1-bit messages through our framework and compare it with two state-of-the-art model-agnostic zero-bit watermarking methods: Soft Red List (Kirchenbauer et al., 2023) and SynthID (Dathathri et al., 2024), which are biased and unbiased methods, respectively. True positive rate (TPR) at 1% false positive rate (FPR) is used as the evaluation metric of this task. The results in Fig. 4 show that BiMark achieves significantly improved performance and comparable detection performance compared to Soft Red List and SynthID, respectively, while maintaining lower perplexity on both Llama3-8B and Qwen2.5 (Team, 2024b).

### 5.2. Text Quality Preservation

Watermarks' impact on generated text quality is assessed in two downstream tasks of language models: text summarization and machine translation. For text summarization, BART-large (Lewis et al., 2019) is employed on the CNN/DailyMail dataset (See et al., 2017), where the performance is evaluated using BERTScore-F1 (Zhang* et al., 2020) and ROUGE-1 (Lin, 2004). For machine translation, MBart (Lewis et al., 2019) is employed on the WMT'16 En-Ro subset (Bojar et al., 2016), where the performance is evaluated using BLEU (Papineni et al., 2002).

*Table 4.* LLM downstream tasks performance.

| Method | Text Summarization | | Machine Translation | |
|---|---|---|---|---|
| | BERTScore | ROUGE-1 | BERTScore | BLEU |
| No Watermark | 32.45±.01 | **38.32**±.02 | **56.21**±.03 | 21.93±.17 |
| Soft Red List (1.0) | 32.11±.02 | 37.97±.04 | 55.74±.18 | 21.36±.16 |
| Soft Red List (1.5) | 31.61±.04 | 37.51±.06 | 55.06±.15 | 20.67±.25 |
| Soft Red List (2.0) | 31.15±.02 | 36.99±.05 | 54.17±.18 | 19.63±.02 |
| Gumbel Sampling | 32.22±.03 | 38.19±.02 | 56.12±.06 | 22.14±.05 |
| $\gamma$-Reweight | 32.25±.02 | 38.09±.02 | 55.67±.05 | 21.49±.14 |
| DiPmark | 32.33±.02 | 38.21±.03 | 56.11±.04 | 21.87±.06 |
| SynthID | 32.45±.03 | **38.32**±.04 | 56.17±.09 | 22.11±.03 |
| BiMark (Ours) | **32.48**±.03 | **38.32**±.03 | 56.14±.07 | **22.15**±.09 |

* "Soft Red List(1.0)" indicates that the value added to green tokens' logit scores are $\delta = 1$. The same applies to "Soft Red List(1.5)" and "Soft Red List(2.0)"

Baseline methods include Soft Red List, Gumbel Sampling (Kuditipudi et al., 2023), $\gamma$-Reweight (Hu et al., 2023), DiPmark (Wu et al., 2023), and SynthID (Dathathri et al., 2024). Soft Red List, SynthID, and BiMark use the same settings as previous experiments, while other methods take default or recommended settings in the original works. In Tab. 4, Soft Red List shows obvious quality degradation, where performance drops significantly as watermark strength increases (indicated by $\delta$). Among unbiased methods, BiMark achieves comparable performance while additionally supporting embedding and extracting useful messages. The consistent performance across both summarization and translation tasks demonstrates that our unbiased multilayer watermarking successfully preserves language models' essential abilities for downstream tasks.

### 5.3. Ablation Study

**Computational cost analysis**. While multilayer reweighting introduces additional computational overhead during token generation, in our experiments, generating a single token using BiMark with a batch size of 1 and 50 takes 0.036s and 0.047s on average, respectively, showing that our BiMark is efficient in parallelized inference.

**Impact of the multilayer mechanism**. An ablation study is conducted across the following four scenarios:

1) *Layer number analysis*: Fig. 5 (a) evaluates watermark detection across varying layer numbers $d = 1, 5, 10, 20$ with fixed $\tilde{\delta} = 1.0$. Results show that detectability initially improves with increasing layers until reaching a peak, then decreases as the number of layers becomes excessive. 2) *Individual layer contribution*: Fig. 5 (b) analyzes each layer's contribution to detection with $\tilde{\delta} = 1.0$ and $d = 10$. Results show that all layers contribute to detection, with shallow layers providing particularly strong signals. 3) *Scaling factor analysis*: Fig. 5 (c) evaluates detection performance with a base scaling factor $\tilde{\delta}$ ranging from 0.1 to 1.0 and fixed $d = 50$. Performance follows the same pattern as layer analysis. 4) *Resilience analysis*: Fig. 5 (d) assesses resilience against 10% word substitution across different layer numbers. Results indicate that watermark resilience improves

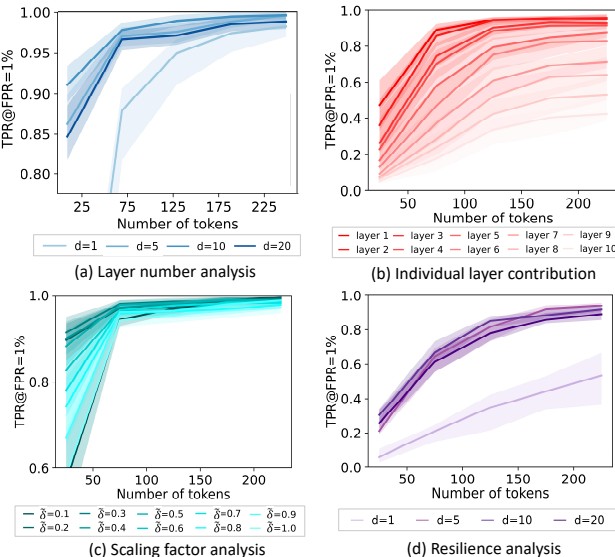

*Figure 5.* Ablation experiments of multilayer reweighting.

with increased layers, slightly decreases after reaching a peak, but remains superior to single-layer approaches.

As shown in App. A.2, watermark detectability depends on the base scaling factor $\tilde{\delta}$ and probability balance between bipartitions. Observation (2) occurs because multilayer reweighting is iterative, with shallow layers having a dominant impact on the final probability distribution. Since shallow layers typically have more evenly distributed probabilities over tokens, their reweighting effects are more substantial and cannot be reversed by subsequent layers. Observation (1) occurs because while adding layers initially strengthens the watermark signal and improves detectability, excessive layers diminish performance since deeper layers contribute minimally to detection while adding noise. Observation (3) occurs because appropriate base scaling factors enable gradual probability reweighting across layers, allowing deeper layers to contribute to detection. Observation (4) occurs because multilayer reweighting creates multiple independent watermark evidence, which enhances the distinguishability between watermarked and clean text.

## 6. Conclusion

This work presents BiMark, a novel watermarking framework for LLMs that simultaneously achieves three critical capabilities: text quality preservation, message embedding capacity, and model-agnostic detection. These properties are essential for practical deployment and are validated through both theoretical and empirical analysis. BiMark's superior message embedding capacity stems from subtle patterns generated by a multilayer mechanism, which reveals new possibilities for exploring more secure watermarking via fine-grained probability distribution reweighting.

## Acknowledgement

This research was partly funded by the Australian Research Council (ARC) under grants FT210100097, DP240101547, DP250102634, and the CSIRO-National Science Foundation (US) AI Research Collaboration Program.

## Impact Statement

This paper presents BiMark, a novel watermarking framework addressing critical challenges in AI-generated content authentication, contributing to responsible LLM deployment. By providing a model-agnostic solution that provides message embedding and text quality preservation, BiMark supports responsible AI governance, contributing to a more transparent and trustworthy AI ecosystem.

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

# A. Proof & Analyses

## A.1. Proof of Theorem 4.2

*Proof.* For any $x \in \mathcal{V}$, we consider $E_e[R_{\theta,e}(P_M)(x)]$ by examining two cases:

Case 1: If $x \in \mathcal{V}_1$,

$$E_e[R_{\theta,e}(P_M)(x)] = \frac{1}{2}[(1+\delta_1)P_M(x)] + \frac{1}{2}[(1-\delta_1)P_M(x)]$$
$$= P_M(x).$$

Case 2: If $x \in \mathcal{V}_0$,

$$E_e[R_{\theta,e}(P_M)(x)] = \frac{1}{2}[(1-\delta_0)P_M(x)] + \frac{1}{2}[(1+\delta_0)P_M(x)]$$
$$= P_M(x).$$

Since this holds for all $x \in \mathcal{V}$, we have $E_e[R_{\theta,e}(P_M)] = P_M$. Note that this result holds regardless of the specific value of $\delta_1$ (and consequently $\delta_0$), as long as they maintain a valid probability distribution. □

## A.2. Type-II Error Analysis of Single-Layer Unbiased Reweighting

To understand the multilayer watermarking scheme, we first analyze the Type-II error rate (false negative rate) of watermark detection in a single-layer reweighting. While the multilayer reweighting involves complex interactions between layers, this single-layer analysis provides crucial insights into the fundamental behavior of the multilayer approach.

**Statistical properties of a single-layer reweighting.** Given a text sequence of length $T$, let $G$ denote the count of tokens belonging to green lists during detection. Under the null hypothesis $H_0$ (i.e., the token sequence contains no watermark), the expected proportion of green tokens is $0.5$ due to balanced vocabulary bipartitions. Under the alternative hypothesis $H_1$ (i.e., the token sequence is watermarked), this proportion deviates from $0.5$ due to the single-layer reweighting scheme. For a single token $x_t$ at position $t$, let $G_t$ be a random variable indicating whether $x_t$ belongs to the corresponding green list.

**Corollary A.1.** *Under a single-layer unbiased reweighting, the expectation and the variance of $G_t$ follow:*

$$E[G_t] = \begin{cases} 0.5 + \tilde{\delta}\tau_t & \text{if } 0 \leq \tau_t \leq \frac{1}{1+\tilde{\delta}}, \\ 1.5 - \tau_t & \text{if } \frac{1}{1+\tilde{\delta}} < \tau_t < 1, \end{cases}$$

$$\text{Var}[G_t] = \begin{cases} 0.25 - \tilde{\delta}^2\tau_t^2 & \text{if } 0 \leq \tau_t \leq \frac{1}{1+\tilde{\delta}}, \\ -0.75 + 2\tau_t - \tau_t^2 & \text{if } \frac{1}{1+\tilde{\delta}} < \tau_t < 1, \end{cases}$$

*where $\tilde{\delta}$ is the base scaling factor for unbiased reweighting.*

*Proof.* Given a vocabulary bipartition $\mathcal{V}_0$ and $\mathcal{V}_1$, and a fair coin flip $e$, in a bit-flip unbiased reweighting, green tokens are defined based on two conditions: when $e = 0$, tokens $x \in \mathcal{V}_0$ are green, and when $e = 1$, tokens $x \in \mathcal{V}_1$ are green. Since the value of $e$ and the vocabulary bipartition are independent, an indicator $G_t$ follows:

$$G_t = \begin{cases} 1 & \text{if } (e = 0 \wedge x \in \mathcal{V}_0) \text{ or } (e = 1 \wedge x \in \mathcal{V}_1), \\ 0 & \text{otherwise.} \end{cases} \tag{4}$$

Let $P_M^t$ and $P_{M,w}^t$ denote the original and reweighted probability distribution at time step $t$, respectively. Note $\tau = \sum_{x \in \mathcal{V}_1} P_M(x)$, according to the definition of bit-flip unbiased reweighting (see Eq. (1)), we have:

$$P_{M,w}(x|x \in \mathcal{V}_1) = \begin{cases} (1+\delta)P_M(x) & \text{if } e = 1 \\ (1-\delta)P_M(x) & \text{if } e = 0 \end{cases}$$

$$P_{M,w}(x|x \in \mathcal{V}_0) = \begin{cases} (1 - \frac{\delta\tau}{1-\tau})P_M(x) & \text{if } e = 1 \\ (1 + \frac{\delta\tau}{1-\tau})P_M(x) & \text{if } e = 0. \end{cases} \tag{5}$$

The expectation of $G_t$ can be derived as follows:

$$
\begin{aligned}
E[G_t] &= \sum_{x \in \mathcal{V}} [P_{M,w}^t(x|x \in \mathcal{V}_1) + P_{M,w}^t(x|x \in \mathcal{V}_0)]G_t \\
&= \sum_{x \in \mathcal{V}} [P_{M,w}^t(x|x \in \mathcal{V}_1, e = 1)P(e = 1) + P_{M,w}^t(x|x \in \mathcal{V}_1, e = 0)P(e = 0) \\
&\quad + P_{M,w}^t(x|x \in \mathcal{V}_0, e = 1)P(e = 1) + P_{M,w}^t(x|x \in \mathcal{V}_0, e = 0)P(e = 0)]G_t \\
&= \sum_{x \in \mathcal{V}} [P_{M,w}^t(x|x \in \mathcal{V}_1, e = 1)P(e = 1) + P_{M,w}^t(x|x \in \mathcal{V}_0, e = 0)P(e = 0)] \qquad (\because Eq.\ (4)) \\
&= 0.5 \sum_{x \in \mathcal{V}} [P_{M,w}^t(x|x \in \mathcal{V}_1, e = 1) + P_{M,w}^t(x|x \in \mathcal{V}_0, e = 0)] \qquad\qquad (\because e \sim \text{Bernoulli}(0.5)) \\
&= 0.5 \sum_{x \in \mathcal{V}} [(1 + \delta)P_M^t(x|x \in \mathcal{V}_1) + (1 + \frac{\delta \tau_t}{1 - \tau_t})P_M^t(x|x \in \mathcal{V}_0)] \qquad\qquad (\because Eq.\ (5)) \\
&= 0.5[(1 + \delta)\tau_t + (1 + \frac{\delta \tau_t}{1 - \tau_t})(1 - \tau_t)] \\
&= 0.5 + \delta \tau_t.
\end{aligned}
$$

Given that $\delta$ is determined by both $\tilde{\delta}$ and $\tau_t$ in Eq. (3), $E(G_t)$ can be expressed as a piecewise function:

$$
E[G_t] = \begin{cases} 0.5 + \tilde{\delta}\tau_t & \text{if } 0 \le \tau_t < \frac{1}{1+\tilde{\delta}}, \\ 1.5 - \tau_t & \text{if } \tau_t \ge \frac{1}{1+\tilde{\delta}}. \end{cases} \tag{6}
$$

To derive $Var[G_t]$, since $G_t$ takes values of either 1 or 0, we have $G_t^2 = G_t$, and consequently $E[G_t^2] = E[G_t]$. Using the variance formula and this property, we have:

$$
\text{Var}[G_t] = E[G_t^2] - E[G_t]^2 = E[G_t] - (E[G_t])^2.
$$

This leads to the following piecewise expression:

$$
\text{Var}[G_t] := \begin{cases} 0.25 - \tilde{\delta}^2 \tau_t^2 & \text{if } 0 \le \tau_t < \frac{1}{1+\tilde{\delta}}, \\ -0.75 + 2\tau_t - \tau_t^2 & \text{if } \tau_t \ge \frac{1}{1+\tilde{\delta}}. \end{cases} \tag{7}
$$

$\square$

**Type-II error analysis.** The Type-II error rate $\beta$ is the probability of failing to detect a watermark when one is present. For the z-test with significance level $\alpha$:

$$
\beta = P(z < z_{1-\alpha}|H_1),
$$

where $z_{1-\alpha}$ is the critical value and $z$ is our test statistic, since the proportion of green lists is 0.5:

$$
z = \frac{G/T - 0.5}{\sqrt{0.25/T}}. \tag{8}
$$

Under $H_1$, the total count $G$ approximately follows a normal distribution by the Central Limit Theorem:

$$
G \sim N(T \cdot E[G_t], T \cdot \text{Var}[G_t]). \tag{9}
$$

Using Eq. (8) and Eq. (9), the expectation and variance of $z$ under $H_1$ can be derived, and under $H_1$, it follows:

$$
z \sim N(2(E[G_t] - 0.5)\sqrt{T}, 4 \cdot \text{Var}[G_t])
$$

Consequently, the Type-II error rate is:

$$
\beta = \Phi(\frac{z_{1-\alpha} - 2(E[G_t] - 0.5)\sqrt{T}}{2\sqrt{\text{Var}[G_t]}})
$$

**Key insights.** The Type-II error rate decreases when $E[G_t]$ increases or $\text{Var}[G_t]$ decreases. When $\tilde{\delta} = 1$, $E[G_t]$ reaches its maximum when $\tau = 0.5$, implying that our watermarking scheme is most effective for high-entropy text, which aligns with previous findings from (Kirchenbauer et al., 2023) and (Kuditipudi et al., 2023). Note that $E[G_t]$ reaches its maximum when $\tau = \frac{1}{1+\tilde{\delta}}$. This actually corresponds to the case when $\tau$ is amplified to occupy the entire probability space. In this case, probabilities in $\mathcal{V}_0$ will be shrunk to $0$. As multilayer reweighting is iterative, this process will concentrate probabilities among a few tokens. This property indicates that shallow layers of reweighting have more impact on the ultimate reweighted probability distribution. Once tokens are shrunk to near-zero probabilities in early layers, even if they are allocated to green lists in deeper layers, their probabilities cannot be effectively recovered.

### A.3. Multilayer Reweighting

We analyze the expectation of reweighted probability distribution layer by layer. Let $P_M^i$ denote the distribution after applying $i$ layers of reweighting. For any layer $i$, we know from Theorem 4.2 that $E_{e^i}[R_{\theta^i, e^i}(P_M^{i-1})] = P_M^{i-1}$. This property holds true at each layer regardless of the outcomes of previous layers due to the fact:

1. Vocabulary bipartitions $(\mathcal{V}_0^i, \mathcal{V}_1^i)$ is independent of $(\mathcal{V}_0^{i+1}, \mathcal{V}_1^{i+1})$;

2. Fair coin flip $e^i$ is independent of $e^{i+1}$;

3. Bipartitions and fair coin flips are independent of each other.

Therefore, by analyzing from the innermost layer outward, we conclude that the entire composition maintains the unbiased property:

$$E_{\boldsymbol{e}}[P_{M,w}] = P_M.$$

### A.4. Property of XOR operaion

We first define our variables. Let $x \in \{0, 1\}$ be our original bit. Let $b \in \{0, 1\}$ be a random bit sampled from Bernoulli$(0.5)$. Let $e = x \oplus b$ be the result of the XOR operation.

We need to prove that $P(e = 1) = 1/2$ and $P(e = 0) = 1/2$, regardless of the value of $x$.

when $x = 0$, $e = 0 \oplus b$. If $b = 1$, then $e = 0$. If $b = 0$, then $e = 1$. Because $P(b = 1) = P(b = 0) = 1/2$, we have $P(e = 0|x = 0) = 1/2$ and $P(e = 1|x = 0) = 1/2$.

When $x = 1$, $e = 1 \oplus b$. If $b = 1$, then $e = 0$. If $b = 0$, then $e = 1$. Because $P(b = 1) = P(b = 0) = 1/2$, we have $P(e = 0|x = 1) = 1/2$ and $P(e = 1|x = 1) = 1/2$.

Since the statement holds true for both possible values of $x$, we can conclude that $e$ follows a Bernoulli$(0.5)$ distribution regardless of the original value of $x$.

# B. More Experiments

To further assess watermark resilience, watermark detection performance against paraphrasing attacks is evaluated. These attacks are performed using DIPPER (Krishna et al., 2023), a fine-tuned language model for watermark evasion with controllable lexical and order diversity parameters.

*Table 5.* Watermark detection against paraphrasing attacks.

| Method | Text Length (Token) | | | | | | | | | | | | | | | |
|---|---|---|---|---|---|---|---|---|---|---|---|---|---|---|---|---|
| | 50 | | | | 100 | | | | 200 | | | | 300 | | | |
| | / | (20,0) | (0,20) | (20,20) | / | (20,0) | (0,20) | (20,20) | / | (20,0) | (0,20) | (20,20) | / | (20,0) | (0,20) | (20,20) |
| Soft Red List | 68.88 | 15.43 | 35.27 | 13.08 | 92.53 | 32.39 | 69.94 | 27.4 | 98.11 | 56.25 | 90.84 | 49.4 | 99.78 | 71.16 | 96.9 | 66.52 |
| SynthID | 97.25 | 54.82 | 87.03 | 50 | 98.04 | 83.14 | 96.52 | 76.13 | 99.48 | 97.83 | 99.55 | 94.75 | 100 | 97.78 | 100 | 97.21 |
| DiPmark | 59.57 | 10.86 | 24.4 | 10.25 | 76.07 | 23.71 | 34.03 | 15.56 | 89.94 | 41.62 | 62.63 | 0.236 | 94.76 | 65.39 | 89.57 | 42.81 |
| BiMark | 97.87 | 67.4 | 89.17 | 59.94 | 98.42 | 78.37 | 95.71 | 70.9 | 99.81 | 91.62 | 99.2 | 87.28 | 100 | 98.93 | 100 | 98.35 |

\* "(20, 0)", "(0,20)", and "(20,20)" follows the notation of (lexical diversity, order diversity).

# C. Algorithms

Alg. 1 summarizes the watermarked text generation process for message embedding discussed in Sec. 4.2. Alg. 2 summarizes the watermark detection process for message extraction discussed in Sec. 4.2. The overall process is visualized in Fig. 6.

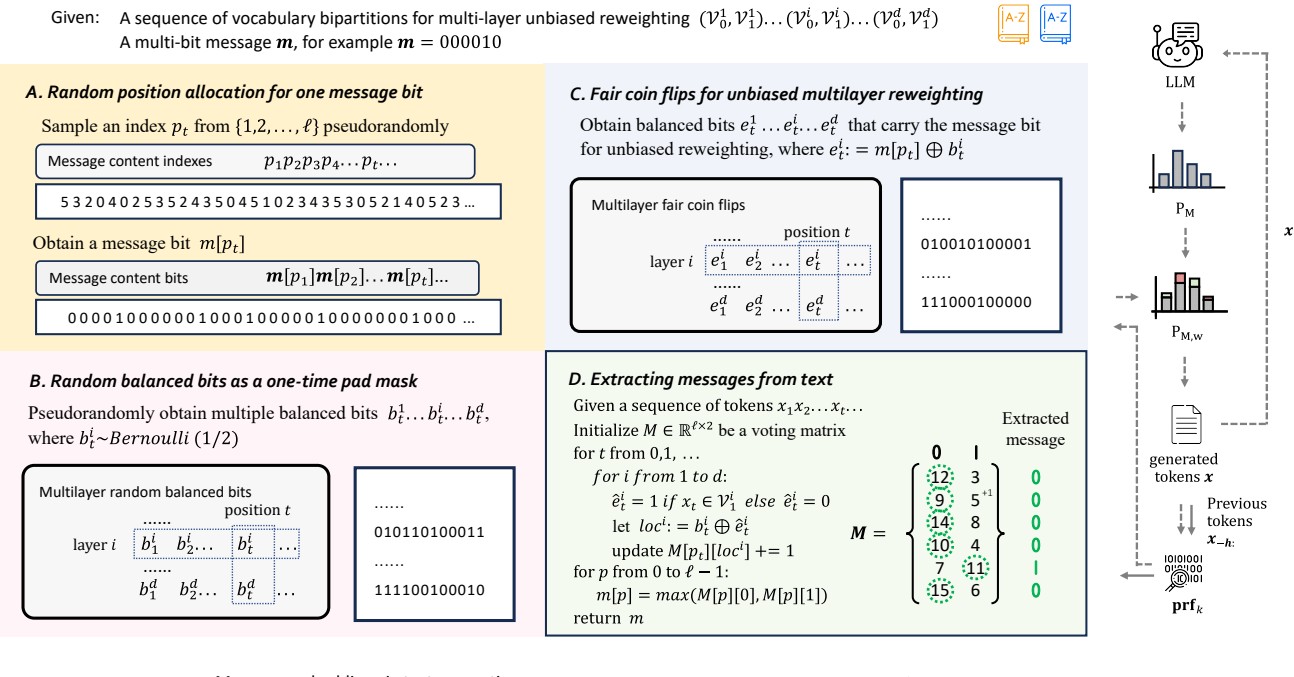

*Figure 6.* The complete process of BiMark. The right part shows original LLM text generation. The watermark embedding process begins when the LLM outputs an original probability distribution $P_M$. Step A pseudorandomly selects a message bit $\boldsymbol{m}[p]$. Step B samples $d$ independent balanced bits $\boldsymbol{b} = [b^1, b^2, \cdots, b^d]$ from Bernoulli(0.5) as a one-time pad mask. Step C applies XOR operations between $\boldsymbol{b}$ and $\boldsymbol{m}[p]$ to obtain fair coin flips $\boldsymbol{e} = [e^1, e^2, \cdots, e^d]$, then conducts multilayer unbiased reweighting. The next token is sampled from the reweighted distribution, and the process continues iteratively. Step D processes token sequences to extract messages. For each token, the message index $p$ and one-time pad mask $\boldsymbol{b}$ are pseudorandomly reconstructed. Based on token subset membership, estimations of fair coin flips $\hat{\boldsymbol{e}}$ are obtained. Using $b^i \oplus \hat{e}^i$, votes for message bits are collected. The final message is extracted via majority voting.

---

**Algorithm 1** Embedding multi-bit message

---

**Input:** a language model LM
        a sequence of tokens as a prompt
        a sequence of balanced vocabulary bipartitions
        $[(\mathcal{V}_0^1, \mathcal{V}_1^1), (\mathcal{V}_0^2, \mathcal{V}_1^2), ..., (\mathcal{V}_0^d, \mathcal{V}_1^d)]$
        a message $\boldsymbol{m} \in \{0,1\}^\ell$, window size $h$
        a base scaling factor $\tilde{\delta}$
        bit-flip unbiased reweighting function $R$
        pseudorandom functions $\mathtt{prf}_\mathrm{p}$, $\mathtt{prf}_\mathrm{b}$

**for** $t = 1, 2, \cdots$ **do**

  1. Apply LM to all prior tokens to get a probability distribution $\mathrm{P_M}^0$ over the vocabulary.

  2. **If** the current previous tokens $\boldsymbol{x}_{-h:}$ in the sliding window have been used as a seed,
     **then** sample a next token $x_t$ from $\mathrm{P_M}^0$;
     **else** record the current previous tokens $\boldsymbol{x}_{-h:}$ and apply $\mathtt{prf}_\mathrm{p}$ and $\mathtt{prf}_\mathrm{b}$ to it to get seed$_\mathrm{p}$ and seed$_\mathrm{b}$.

  3. Sample an index $p_t \in \{1, 2, \cdots, \ell\}$ using seed$_\mathrm{p}$.
     Sample $b_t^1, b_t^2, \cdots, b_t^d \sim \mathrm{Bern}(0.5)$ using seed$_\mathrm{b}$.

  **for** $i = 1, 2, \cdots, d$ **do**
     4. Let $\theta_i = (\mathcal{V}_0^i, \mathcal{V}_1^i, \tilde{\delta})$, and calculate

$$e_t^i = \boldsymbol{m}[p_t] \oplus b_t^i.$$

     5. Obtain the $i$-layer reweighted distribution

$$P_M^i = R_{\theta_i, e_t^i}(P_M^{i-1}).$$

  **end for**

  6. Sample the next token $x_t$ from $P_{M,w} = P_M^d$.

**end for**

---

**Algorithm 2** Extracting multi-bit message from text

---

**Input:** a sequence of balanced vocabulary bipartitions
        $[(\mathcal{V}_0^1, \mathcal{V}_1^1), (\mathcal{V}_0^2, \mathcal{V}_1^2), ..., (\mathcal{V}_0^d, \mathcal{V}_1^d)]$
        message length $\ell$, window size $h$
        pseudorandom functions $\mathtt{prf}_\mathrm{p}$, $\mathtt{prf}_\mathrm{b}$

1. Initialize a $\ell \times 2$ voting matrix $\boldsymbol{M}$

**for** $t = 1, 2, \cdots$ **do**

  2. **If** the current previous tokens $\boldsymbol{x}_{-h:}$ in a sliding window $h$ have been used as a seed,
     **then** skip the current token and continue;
     **else** record current previous tokens $\boldsymbol{x}_{-h:}$ and apply it to $\mathtt{prf}_\mathrm{p}$ and $\mathtt{prf}_\mathrm{b}$ to get seed$_\mathrm{p}$ and seed$_\mathrm{b}$.

  3. Sample an index $p \in \{1, 2, \cdots, \ell\}$ using seed$_\mathrm{p}$.
     Sample $b^1, b^2, \cdots, b^d \sim \mathrm{Bern}(0.5)$ using seed$_\mathrm{b}$.

  **for** $i = 1, 2, \cdots d$ **do**
     4. $\hat{e}_t^i = 1$ **if** $x_t \in \mathcal{V}_0^i$ **else** $\hat{e}_t^i = 0$.
     5. $loc_t^i = b_t^i \oplus \hat{e}_t$.
     6. update $\boldsymbol{M}[p_t][loc^i]$ += 1.
  **end for**

**end for**

**for** $p$ from 1 to $\ell$ **do**
     7. $\boldsymbol{m}[p] = \max(\boldsymbol{M}[p][0], [p][1])$
**end for**

8. Return the extracted message $\boldsymbol{m}$

---

## D. Examples of Watermarked Text

| | Text | Detection |
|---|---|---|
| Prompt | A food processor opens up a world of possibilities, such as hot soups or icy cold concoctions. It may be a bit bulky, but processors can be used to make raw desserts, banana ice cream and pie crusts. For more savory options, processors are great for quickly ricing and shredding vegetables. Get your own for less than $30 on Amazon. 2. Wooden spoons are going to be your new best friends. Metal tools scratch many pots and pans, but wood won't cause them any harm. | N/A |
| No Watermark | Try making banana ice cream with only an immersion blender instead of a traditional ice cream maker and a food processor. It's a fast and no-mess way to make a tasty treat. Try this Cinnamon & Honey Banana Nice Cream Recipe: 3 ripe frozen bananas 1 tbsp. cinnamon 1 cup almond milk Blend in a large food processor or blender with an immersion mixer until smooth. Top with honey, sliced almonds, or your favorite toppings and enjoy! 3. A good blender will work wonders for smoothies and soup purees and won't cause you to break your wallet. If you are an ingredient combo junkie, a blender may be a more economical option than a processor. You can blend together a variety of foods in your blender to make raw vegan desserts like cashew or macadamia nut-based creams. 4. If you think you can forego ice cream, think again! You can make raw ice cream alternatives that are healthy for your waistline but still taste great. Try this recipe fora Raw Coconut Ice Cream: 4 tbsp almond milk ¼ tbsp vanilla extract ½ cup unsweetened shredded coconut Blend in your food processor and pour over chopped bananas on a plate. Freeze for 15-20 minutes. Enjoy! 5. Spaghetti squash is a must have for any raw vegan. Not only is spaghetti squash delicious, but they can be very filling and require no pasta machine. Spaghetti squash are also very versatile and can be eaten in a variety of ways. 6. A dehydrator is another great raw food tool. Try making raw vegan jerky, dehydrating fruit and veggies to make crispy chips or making desserts like cookies or brownies. 7. You can make your own veggie chips. 8. You can even make your own veggie chips. They aren't bad, so stop buying them! Use your dehydrator and some non-stick parchment sheets to make your own at home in three simple steps. Thinly slice vegetables (thinness depends on crispiness), dehydrate for 12 hours and enjoy! 9. You can add a little spice to any recipe with a small jar containing chili flakes. They come in handy for many dishes. From savory salads made with freshingredients to simple avocado toast, they make any dish a little bit more interesting. 10. Another great way to add some flavor to your raw cuisine is through the use of spices. Spices can be a fun and easy way to boost flavor. A good salt-free blend | z-score: 0.9758

p-value: 0.1645

N/A |
| BiMark | They're also much easier to clean: Just wash it with soap and water, and leave it out in the open air to dry. You should also invest in a silicone spoon. They're great for scraping bowls, stirring sauces or blending sauces into a puree. 3. Use a mesh sieve to strain your sauce or pasta water. You don't need to buy one — simply purchase a vegetable steamer or cheesecloth from your local supermarket. Lay the steamer or cloth in a heat-safe strainer, and pour the sauce through it. Alternatively, try tying up the steamer or cloth around a wooden spoon and straining that way. 4. If you're working with a tiny kitchen and limited counter space, consider investing in a collapsible silicone spatula. Like all high-quality tools, it's heat resistant and dishwasher safe. Silicone spatulas are great for stirring and mixing, because the soft rubber won't scratch any cookware. They also have flat edges, which make them perfect for scraping those last bits of batter out of bowls. 5. Don't buy tongs or a fork for flipping food. A spatula makes the whole business much easier. Spatulas are incredibly adaptable — they work on oiled and greased surfaces, and they're ideal for tossing or turning the salad. The handle is flat, which means the spatula is safe to put on a chopping board. Unlike a fork or slotted spoon, the spatula is gentle on fragile fish and doesn't disturb the delicate bits of salad leaves. 6. For less than $30 you can buy a bamboo cutting board. Be sure to select a board made of untreated bamboo, and purchase one with a groove running down the center for easy food disposal. Since the bamboo's naturally water resistant, there's no need to soak the board in water or bleach; you can just wash it with hot water and soap. Be careful, though: Wooden cutting boards can crack if they get too hot, so keep it off the stove. 7. Don't throw out your coffee grinder. It's an ideal gadget for transforming spices into powdery blends. Keep an eye on the grind: If it gets too fine, the grinder may release static electricity, which will make it hard for spices to exit the chamber. If they keep getting stuck, mix the spice around with a spoon. For even more flavor, try placing the grinder on an electric heater. If you're using whole spices, let them heat up directly. This will help open up | z-score: 17.5733

p-value: 1.97E-69

extraction message:
"1111010111100010100110001100011"

voting matrix $M=$
[[24, 96], [65, 79], [44, 84],[30, 58],
[73, 47],[32, 48],[64, 40], [79, 113],
[33, 78], [24, 56], [30, 74], [57, 39],
[98, 46], [65, 39], [49, 79], [54, 26],
[48, 87], [49, 31], [44. 12], [67, 109],
[30, 42], [13, 3], [63, 17], [38, 34],
[36, 68], [126, 162], [90, 38], [93, 51],
[81, 39], [46, 90], [15, 49], [32, 71]]

message extraction rate: 1.0 |

| | Text | Detection |
|---|---|---|
| Prompt | Before SAP Labs designed its new digs in Palo Alto, company executives asked employees how to make it a welcoming place to work. After all, that's where they spend most of their day. The answer? Windows. Lots of windows letting in lots of light. The German-based business software company complied – even going so far as to move executives away from windows to give more workers access to daylight. As the country embraces "green" building design – be it in the workplace or at home – daylight has become a premium. | N/A |
| No Watermark | "Light has so many things to contribute to our satisfaction and productivity levels," said Mark de la Vergne, president of the International Commission on Illumination. "We're just coming to realize that." Daylight not only promotes healthier lives and work environments, but can be an economic incentive for businesses and homeowners and a tool for the real estate market. It can boost productivity, stimulate better decision-making and reduce energy and operating costs – all reasons green builders are encouraging more natural light than in past years. "Daylight is a commodity that we've neglected for so long as Americans," said David Johnston, president of Pacific Green Building and an expert on designing green buildings. "I think we're starting to see a real awakening." Green design is a trend that has swept over the Western U.S. in the last decade. As more people seek efficient, earth-friendly homes, they demand natural light, especially for indoor activities. "The value of natural light is huge," said Tim Eddy, senior principal for architecture and design company ZGF. "With the amount of time we spend indoors, it's the way for all of us to remain healthfully connected to a natural cycle." People naturally respond to direct sunlight. It's been part of our environment forever. It affects us from the inside out – the way we feel, how well we sleep, how much energy we have, and how we think. "There is an energy in the air, and you can just get a spark from it," Eddy said. Natural light is a human response to nature, de la Vergne said. It's also an economic response to a growing building trend. In the last decade, green building has taken its place alongside the more practical – and often more costly – considerations of building design and construction. It's considered a "must" item for homeowners, office buildings and residential builders throughout the country. The U.S. Green Building Council, which established a rating standard for green building, now provides certification for tens of thousands of buildings – residential and commercial – across the country. There aren't statistics on how many homes have "natural light" on their wish list. But "green" homes are increasingly popular. The Leadership in Energy and Environmental Design certification program – popular for homes, particularly in the West, and a requirement for green buildings owned by the federal government – certifies residential buildings based on criteria such as energy efficiency, water efficiency, location, material selection and sustainable design. "Any homebuyer who plans to stay in a home longer than five years is better | z-score:0.9899

p-value:0.1611

N/A |
| BiMark | When the federal government set new energy and environmental requirements this summer, it raised standards for indoor lighting, calling for buildings to let in more natural light through windows. That goal dovetails nicely with energy efficiency goals set in recent years. "Daylight is free, and at a time when energy is super expensive, you can save money and reduce carbon emissions," said Michael Howard, the new director of lighting research at the Pacific Northwest National Laboratory in Richland. Howard says the new research, including the first-ever test bed to be constructed to measure natural light, could lead to big energy savings in the future. Lighting represents the largest share of electricity use in commercial buildings nationwide – more than the air conditioners and furnaces in buildings combined. And lighting accounts for 27 percent of the power consumed by homes. The new energy and environmental requirements being considered by Congress would lead to big improvements in energy efficiency. At the same time, a coalition ofexperts at the Pacific Northwest National Laboratory is looking at the impact of natural light on worker productivity and job satisfaction. This all has become important in the new age of energy efficiency. "Green building is the buzz now," Howard said. "Green building is the buzz now." How much sunlight do we need? One of the questions the new research, including the test bed to be constructed next year, is what is the right amount of light to set the stage for a productive work environment. Howard says most federal buildings aim for lighting that would equal the equivalent of a full moon. But lighting that would equal the equivalent of a full moon is actually too much light. In fact, he says it could cause problems with worker productivity. "Some studies are showing that you need less light, about two to three times less, to be more productive," he said. The new test bed to be constructed next year is a state-of-the-art 25,000-square-foot office building at Richland's Hanford site. It will study everything from window placements to the impact of natural light on worker productivity. the new test bed will look for ways to better design natural light to maximize worker productivity. "We think that it'll be very helpful in understanding if we're really getting the benefits," he said. The new test bed to be constructed next year is a state-of-the-art 25,000-square-foot office building at Richland's Hanford site. It will study everything from window placements to the impact of natural light on worker productivity. The new test bed will look for ways to better design natural light to maximize worker | z-score:20.47

p-value: 1.89e-93

extracted message:
"1010000011001101000110000010100"

voting matrix $M=$
[[26, 78], [23, 17], [23, 57], [105, 55], [63, 41], [108, 68], [115, 53], [134, 74], [52, 100], [18, 54], [117, 43], [53, 19], [35, 93], [51, 85], [85, 43], [57, 87], [84, 52], [67, 21], [58, 46], [24, 56], [53, 91], [67, 53], [67, 29], [116, 60], [62, 34], [77.6, 66.4], [57, 15], [25, 47], [65, 31], [62, 146], [137, 63], [92, 44]]

message extraction rate: 1.0 |