# OpenReview forum: "BiMark: Unbiased Multilayer Watermarking for Large Language Models"
_ICML.cc/2025/Conference — ICML 2025 poster_

### Official Review · Reviewer_jksy · 2025-03-10

**Overall Recommendation:** 3

**Summary:**

In this work, the authors introduce BiMark, which encodes and extracts multi-bit messages via watermarking while preserving text quality. Meanwhile, BiMark enables model-agnostic detection via a bit-flip unbiased reweighting and enhances detectability via a multi-layer architecture.

**Claims And Evidence:**

The claims are clear and reasonable, yet lack the explicit correspondence of evidence.

**Essential References Not Discussed:**

N/A

**Experimental Designs Or Analyses:**

Firstly, the experiments are not fairly compared. The authors evaluate the model-agnostic detection of BiMark by comparing SynthID and KGW (the author mentioned only shot red list in analysis), while other representative works are missing, e.g., soft red-list, MPAC, and GINSEW.

Secondly, the ablation studies are not clear in presentation or analyses. It is hard to identify the legend correspondence, and it is confusing to take d=50 for studying the detectability, yet d=50 is not studied in the effect of multilayer.

The analyses are not organized in a readable way, making it doubtful to use generative language models directly.

**Methods And Evaluation Criteria:**

BiMark enables model-agnostic detection by a vocabulary size-based reweighting approach and enhances detectability via multi-layer reweighting mechanism. However, the main idea of preserving text quality is borrowed from unbiased reweighting.

**Other Comments Or Suggestions:**

1 The experimental analysis should enhance
2 The figure could be illustrated more clearly.
3 The project should release the code for better contribution.

**Other Strengths And Weaknesses:**

Strenghts
1 The authors provide a clear and comprehensive category of related watermarking work.
2 The motivation is interesting and worth exploring.
3 BiMarker is a model-agnostic approach that is compatible with existing generative models.



weakness:
1 Both Fig.1 and Fig.6 need more illustration. It is hard to follow by the current version.
2 The presentation is hard to follow and needs to be polished better.
3 The computation burden requires deeper analysis and potential solutions.
4 BiMark is vulnerable to attacks, which needs deeper exploration.

**Questions For Authors:**

See Above

**Relation To Broader Scientific Literature:**

BiMark's contributions are situated within the broader scientific literature on watermarking techniques for Large Language Models (LLMs). Specifically, BiMark integrates multilayer architecture enhancing detectability without compromising generation quality, and an information encoding approach supporting multi-bit watermarking.
BiMark advances the field of digital watermarking, offering a robust solution for content authentication and intellectual property protection in AI-generated text.​

**Theoretical Claims:**

Firstly, it needs to provide computation complexity of BiMark.
Secondly, the authors provide Type-II error of single-layer unbiased reweighting, while failing to extend to multilayers.

---

> ### Author Rebuttal · Authors · 2025-04-01
>
> **Q1**:*Computational cost*
>
> **R1**:
> For a single-layer reweighting, the method requires first calculating the probability of a vocabulary subset, then obtaining scaling factors based on these original probabilities, and finally adjusting the probability of candidate tokens. The complexity of this operation is $O(|\mathcal{V}|)$.
>
> For multi-layer setting, the operation is iterative, making the computation cost grow linearly with the increase in the number of layers, leading to a total complexity of $O(|d\mathcal{V}|)$.
>
> The table below shows the runtime for BiMark during a single inference pass in our experiments:
>
> |Method|Batch_size=1|Batch_size=50|
> |---------------------|-----------|--------|
> |BiMark|0.036 s|0.047 s|
>
> The experimental result shows BiMark can benefit significantly from large batch size.
>
> **Q2**: *Type-II error of unbiased multi-layer reweighting.*
>
> **R2**: Although it is challenging to provide a close-form expression of type-II error rate of multi-layer watermarking at present since the current layer's probability depends on the reweighted distribution of the previous layer, the analytical idea is the same as single-layer reweighting.
> Specifically, the detectability of multi-layer watermarking gradually decreases as unbiased reweighting reduces distribution entropy across layers.
>
> Intuitively, our method iteratively adjusts probability on tokens through unbiased reweighting based on our red-green vocabulary partitions. In a simplified case with two probabilities $p$ and $1-p$, each layer of reweighting transfers some probability $\Delta$ between these groups with fair chance, creating a detectable pattern.
>
> According to type-II error rate analysis of our single-layer unbiased reweighting, higher entropy distributions allow higher expected green ratio in watermarked text, improving detectability.
> However, this process also inherently reduces entropy because entropy $(-\sum_x p(x)\log p(x))$ is concave, and our reweighting creates a more "peaked" distribution in expectation.
>
> As a result, with each successive reweighting layer, the entropy of the reweighted distribution decreases in expectation, leading to diminishing marginal contributions to detectability.
>
> This analysis explains our empirical observation that detection performance does not increase monotonically with the number of layers. Deeper layers operate on increasingly low-entropy distributions, offering proportionally smaller gains in overall watermark strength.
>
> **Q3**: *Experimental comparison.*
>
> **R3**: We would like to clarify our experimental choices:
>
> For zero-bit case, BiMark was compared with Soft-Red-List (it only supports zero-bit watermarking). The labelling of Soft-Red-List as "KGW" in the figures will be corrected for consistency in the revised version.
>
> For multi-bit case, BiMark was compared with MPAC. MPAC is not available for the zero-bit scenario since it is specifically designed for multi-bit watermarking tasks and is built upon Soft-Red-List.
>
> GINSEW \[1\] is not available as it is intended for a different purpose—specifically, protecting text-generation models from theft through distillation rather than detecting watermarks in individual text segments.
>
> \[1\] Protecting language generation models via invisible watermarking.
>
>
> **Q4**: *Ablation study*
>
> **R4**: In Fig.5(c\), we set the number of reweighting layers to $d=50$. We apologize that the legend does not clearly reflect this setting.
> We chose this large value to more clearly demonstrate how scaling factor affects watermark detectability when the number of layers is fixed.
> While the relationship between scaling factor and detectability holds for any value of $d$, the effects are more obvious with larger $d$ values, making the trends easier to visualize.
>
> **Q5**: *Potential vulnerability to attack*
>
> **R5**:
> We have additionally evaluated the vulnerability of our method under paraphrasing attack, as shown in R3 to Reviewer kPVR.
> Our current work focuses primarily on the watermarking mechanism itself. We acknowledge the importance of  attack resistance, and we will investigate broader attack scenarios and conduct vulnerability analysis as part of our future work.
>
> **Q6**: *Code release*
>
> **R6**: Thank you for suggestions. We plan to release the code implementation at the time of the final decision announcement.
>
> **Q7**:*Concerns about representation*
>
> **R7**: Thank you for your suggestions. We will enhance our paper by: (1) providing more detailed experimental analysis (2) redesigning Figs. 1 and 6 to better illustrate BiMark's mechanisms; and (3) improving overall presentation with consistent terminology and clearer explanations throughout the paper.

---

> > ### Comment · Reviewer_jksy · 2025-04-04
> >
> > Thank you for your response. It mitigates most of my concerns. Considering the potentially unresolved issues, I would like to keep the positive rating unchanged.

---

### Official Review · Reviewer_oDDU · 2025-03-10

**Overall Recommendation:** 4

**Summary:**

The paper presents BiMark, a watermarking technique for large language models (LLMs) that ensures text quality preservation, model-agnostic detection, and message embedding capacity—three key properties often challenging to balance in existing watermarking methods. BiMark introduces a bit-flip unbiased reweighting mechanism that enables detection without model access, a multilayer reweighting approach to enhance detectability, and an information encoding scheme supporting multi-bit watermarks. Through theoretical analysis and empirical validation, BiMark achieves higher extraction rates for short texts while maintaining low perplexity and comparable performance to non-watermarked text in downstream tasks. The framework provides a robust, practical, and efficient watermarking solution for AI-generated content detection.

**Claims And Evidence:**

BiMark is evaluated for text quality preservation, model-agnostic detection, multi-bit embedding, and robustness against text modifications. Empirical results indicate that unbiased reweighting minimizes impact on text fluency and perplexity, with watermark detection verified on Llama3-8B and Qwen2.5-3B. Message embedding is tested through extraction rate comparisons (Table 2), showing up to 30% higher recovery for short texts. Robustness is assessed using synonym substitution experiments (Table 3), though further evaluation against paraphrasing and other adversarial attacks could enhance understanding of the method’s resilience. These results contribute to evaluating BiMark’s effectiveness across different aspects of LLM watermarking.

**Essential References Not Discussed:**

The references are sufficient to the best of the reviewer’s knowledge, covering key works on LLM watermarking, unbiased reweighting, and robustness.

**Experimental Designs Or Analyses:**

The paper evaluates message embedding capacity by comparing BiMark’s extraction rates with state-of-the-art methods (MPAC). Results (Table 2) indicate higher extraction rates and lower perplexity. Robustness is tested using synonym substitution, but further evaluation against paraphrasing or adversarial rewriting is needed. Text quality preservation is assessed with summarization and translation tasks (CNN/DailyMail, WMT’16), using BERTScore, ROUGE, and BLEU (Table 4).

**Methods And Evaluation Criteria:**

(1) Effectiveness and Robustness: BiMark is evaluated on multiple models (Llama3-8B, Qwen2.5-3B) and NLP tasks (summarization, translation). Text quality preservation and model-agnostic detection are assessed using perplexity and performance metrics, while message embedding capacity is examined through extraction rate comparisons.

(2) Practicality and Efficiency: The selected datasets align with watermarking evaluation needs. The paper does not explicitly analyze the computational efficiency of multilayer reweighting or discuss detailed strategies for large-scale deployment, which could provide further clarity on real-world applicability.

**Other Comments Or Suggestions:**

Refer to the comments

**Other Strengths And Weaknesses:**

Strong points

The paper presents an approach to LLM watermarking using unbiased reweighting and multilayer detection, aiming to balance text quality, model-agnostic detection, and message embedding. While prior methods often degrade text fluency for watermark robustness, BiMark seeks a trade-off to maintain detectability without significant quality loss. Theoretical foundations, including unbiased reweighting proofs and type-II error analysis, support its claims. Empirical validation on Llama3-8B and Qwen2.5-3B in summarization and translation tasks demonstrates its effectiveness. Its model-agnostic design removes dependence on internal parameters, suggesting scalability for AI-generated content authentication, though real-world feasibility remains an open question.

Weak points

(1)While the paper tests robustness against synonym substitution, it does not evaluate more sophisticated adversarial attacks, such as paraphrasing-based obfuscation, GAN-based perturbations, or adaptive attacks designed to break watermark detection. The authors are encouraged to evaluate the proposed method against other attacks or discuss the limitations of the watermarking approach.

(2)The paper does not provide explicit inference-time benchmarks for the multilayer reweighting method, making it unclear how computational overhead scales with increasing layers or longer text sequences. The authors are suggested to report the runtime or discuss the computational complexity of the proposed method.

(3)The multilayer watermarking approach shows non-monotonic performance as the number of layers increases. However, the paper does not explore adaptive selection strategies that could optimize performance for different text lengths or generation settings. The authors are encouraged to provide explanations for this observation.

(4)Vague symbol and statements. The notation M[i][v] is used in the multi-bit watermark extraction process. However, it is not immediately clear whether M stores absolute counts or normalized probabilities, making it difficult to assess how well the voting mechanism works under noisy conditions. Moreover, it says that “However, the detection performance does not improve monotonically with the number of layers” While the paper mentions this non-monotonic behavior, it does not clearly explain why it occurs or whether it is due to statistical noise, overfitting, or implementation constraints.

**Questions For Authors:**

Refer to the comments

**Relation To Broader Scientific Literature:**

BiMark builds on prior LLM watermarking techniques like Soft Red-List (Kirchenbauer et al., 2023) and MPAC (Yoo et al., 2023b) but improves text quality preservation, model-agnostic detection, and multi-bit embedding. It extends unbiased reweighting (Hu et al., 2023) with a multilayer approach, enhancing watermark robustness while maintaining detection reliability without model access.

**Theoretical Claims:**

The proof for unbiased reweighting (Theorem 4.2) demonstrates that probability distributions remain unchanged, ensuring unbiasedness across multiple layers through statistical independence. The reasoning follows standard probability principles, with no major errors identified. The type-II error analysis applies the Central Limit Theorem and z-tests, though additional empirical validation under different entropy conditions could improve reliability.

---

> ### Author Rebuttal · Authors · 2025-04-01
>
> **Q1**: Sophisticated adversarial attacks
>
> **R1**: We conducted advanced paraphrasing attacks and reported experimental results in our response to Reviewer kPVR in *R3*.
>
> **Q2**: Inference-time benchmarks
>
> **R2**: We analyzed the computational cost and give experimental time cost of BiMark. Please refer to our response to Reviewer jksy in *R1*.
>
> **Q3**: Non-monotonic performance as the number of layers increases
>
> **R3**: The non-monotonic relationship between layer count and detection performance can be explained through our analytical framework:
>
> - Initial enhancement: Adding layers initially strengthens detection by creating more statistical patterns across independent vocabulary partitions.
> - Entropy reduction effect: As shown in our theoretical analysis (*R2* of Our response to Reviewer jksy), each layer of unbiased reweighting reduces the expected entropy of the probability distribution.
> - Diminishing benefits mechanism: When too many layers are applied, the probability distribution becomes increasingly concentrated, causing later layers to operate on highly skewed distributions. This creates a situation where green lists in later layers may have extremely low probability, making generated tokens unlikely to fall into these green lists and adding noise to the detection process.
>
> We observed this phenomenon more prominently with larger scaling factors, as they cause more dramatic probability redistribution at each layer, accelerating both the initial enhancement and subsequent diminishing benefits.
>
> **Q4**: Meaning of symbols
>
> **R4**: $M$ stores absolute counts for detection.
> During detecting watermark, a ratio of green counts is calculated based on the absolute counts in $M$.
> Though there is noise caused by randomness of text generation, for certain length of watermarked text, the green ratio of watermarked text will significantly exceed 1/2 and that of non-watermarked text will be around 1/2.

---

### Official Review · Reviewer_kPVR · 2025-03-10

**Overall Recommendation:** 3

**Summary:**

In this paper, the authors introduces BiMark, a watermarking framework for LLMs designed to address the challenges of text quality preservation and model-agnostic detection. BiMark utilizes a bit-flip unbiased reweighting mechanism, a multi-layer architecture, and an advanced information encoding strategy to embed watermarks without degrading text quality. Through theoretical and experimental validation, BiMark maintains comparable performance on downstream tasks such as summarization and translation.

**Claims And Evidence:**

In abstract, the authors claimed that "through theoretical analysis and extensive experiments, we validate that, compared to state-of-the-art methods, BiMark reducing perplexity by up to 11%." However, the unbiased property only ensures that the distribution of watermarked LMs is equal to the original LM distribution. Reducing the PPL should not be expected in an unbiased watermark. Thus, this experimental observation contradicted with the unbiased property introduced in this paper.

In introduction, the authors claimed that "adapting existing unbiased reweighting methods for reliable model-agnostic detection is challenging." However, this challenge has been addressed by the prior work e.g. dipmark and SynthID-Text [1]. Thus, the model-agnostic detection is generally not a challenge in LLM watermarking research.

[1] Scalable watermarking for identifying large language model outputs. Nature, 2024.

**Essential References Not Discussed:**

Gamma-reweight and dipmark are cited in the work, but the authors fail to discuss those two methods, given the similarity between the bit-flip reweight and the prior work.

**Experimental Designs Or Analyses:**

In prior work, the watermark algorithms are usually evaluated from three perspective: detectability, quality, and robustness. This paper only consider the detectability and the quality, a comprehensive evaluation of the robustness of the proposed method is missing.

**Methods And Evaluation Criteria:**

This work missing several important baseline watermarking methods. In multi-bit watermark detection experiments,  Qu et al., 2024 [1] and Fernandez et al., 2023 [2] should be included as baseline. In zero-bit distortion free watermark detection experiments, unbiased watermark and dipmark should also be included.

Besides, the robustness of the proposed watermark is not measured in the experiments.

[1] Provably Robust Multibit Watermarking for AI-generated Text via Error Correction Code.

[2] Three bricks to consolidate watermarks for large language models.

**Other Comments Or Suggestions:**

No

**Other Strengths And Weaknesses:**

See above comments

**Questions For Authors:**

No

**Relation To Broader Scientific Literature:**

The reweight algorithm proposed in this paper is similar to the gamma-reweight and the dipmark. The authors should compare their method with the prior work and discuss why the proposed method is better than the prior work.

**Theoretical Claims:**

The theoretical claims look correct to me

---

> ### Author Rebuttal · Authors · 2025-04-01
>
> **Q1**:*Perplexity*
>
> **R1**:
> We reported perplexity in our experiments because it is one of the most commonly used metrics for evaluating the quality of generated text. We will revise the relevant statement in the final paper.
>
> **Q2**:*Challenge of unbiased watermark and model-agnostic detection*
>
> **R2**:
> We agree with you that prior work such as DipMark and SynthID has addressed the model-agnostic requirement. However, our work uniquely aims to integrate three critical properties—text quality preservation, model-agnostic detection, and message embedding capacity—into a single cohesive framework, a task that remains challenging in the current research landscape. We will revise our statement accordingly to prevent confusion.
>
>
> **Q3**:*Baseline methods and evaluation criteria*
>
> **R3**:
> Additional comparative evaluations with \[1\] are conducted, as presented in the following table. Note that \[2\] is not suitable as a baseline due to its significant inefficiency. Previous studies have confirmed its high computational cost—for example, embedding a 32-bit message requires approximately 29,000 seconds \[1\].
>
> The table below reports the bit match rates of multi-bit watermark detection experiments on 8-bit, 16-bit and 32-bit message embedding and extraction, where Length indicates the number of tokens in the watermarked text being detected.
> |Length|50|50|50|100|100|100|200|200|200|300|300|300|
> |-------------------------|:-----:|:------:|:------:|:-----:|:------:|:------:|:-----:|:------:|:------:|:-----:|:------:|:------:|
> |Method|8-bit|16-bit|32-bit|8-bit|16-bit|32-bit|8-bit|16-bit|32-bit|8-bit|16-bit|32-bit|
> |MPAC\[3\]|78.81|66.06|51.03|89.75|78.21|65.35|96.4|89.04|78.15|98.57|93.83|84.7|
> |BCH\[1\]|96.54|71.94|49.08|98.74|84.4|64.67|99.16|95.25|85.46|100|96.21|90.46|
> |BiMark|95.26|85.55|66.35|97.62|93.31|82.69|98.15|95.54|89.68|97.88|95.86|90.22|
>
> Compared with \[1\], BiMark is an unbiased watermarking method, and the Error Correction Code component of \[1\] has the potential to integrate with BiMark to further improve multi-bit watermark performance.
>
> Additional comparative evaluations with DiPmark \[4\] are conducted.
> We exclude comparison with Gamma-Reweight \[5\] as it is not model-agnostic detectable, thus beyond the scope of the application scenario of our study.
> Additional evaluations on robustness are conducted through the advanced paraphraser model Dipper \[6\].
> The table below reports TPR@1%FPR of watermark detection on watermarked text with and without paraphrasing, where (20,0) indicates parameters lex_diversity and order_diversity of Dipper are 20 and 0.
> |Length|50|50|50|50|100|100|100|100|200|200|200|200|300|300|300|300|
> |---------------------------------------------|:---------:|:-------:|:-------:|:--------:|:---------:|:-------:|:-------:|:--------:|:---------:|:-------:|:-------:|:--------:|:---------:|:-------:|:-------:|:--------:|
> |Method|-|(20,0)|(0,20)|(20,20)|-|(20,0)|(0,20)|(20,20)|-|(20,0)|(0,20)|(20,20)|- |(20,0)|(0,20)|(20,20)|
> |Soft-Red-List\[7\]|68.88|15.43|35.27|13.08|92.53|32.39|69.94|27.4|98.11|56.25|90.84|49.4|99.78|71.16|96.9|66.52|
> |SynthID\[8\]|97.25|54.82|87.03|50|98.04|83.14|96.52|76.13|99.48|97.83|99.55|94.75|100|97.78|100|97.21|
> |DiPmark\[4\]|59.57|10.86|24.4|10.25|76.07|23.71|34.03|15.56|89.94|41.62|62.63|0.236|94.76|65.39|89.57|42.81|
> |BiMark|97.87|67.4|89.17|59.94|98.42|78.37|95.71|70.9|99.81|91.62|99.2|87.28|100|98.93|100|98.35|
>
> \[1\] Provably Robust Multibit Watermarking for AI-generated Text via Error Correction Code.
>
> \[2\] Three bricks to consolidate watermarks for large language models.
>
> \[3\] Advancing Beyond Identification: Multi-bit Watermark for Large Language Models.
>
> \[4\] A Resilient and Accessible Distribution-Preserving Watermark for Large Language Models.
>
> \[5\] Unbiased Watermark for Large Language Models.
>
> \[6\] Paraphrasing evades detectors of ai-generated text, but retrieval is an effective defence.
>
> \[7\] A Watermark for Large Language Models.
>
> \[8\] Scalable watermarking for identifying large language model outputs.
>
> **Q4**: Robustness analysis and evaluation
>
> **R4**: A primary robustness evaluation has been provided in Section 5.1 Table 3 in our paper. **Additional evaluations are conducted to test BiMark's resilience against paraphrasing attacks, as outlined in R3**.
>
>
> **Q5**:Comparison with DiPmark and Gamma-Reweight
>
> **R5**:
>
> |Feature / Method| DiPmark | Gamma-Reweight | BiMark |
> |------------------------------------------------------------------------------|---------|----------------|--------|
> Multi-layer mechanism for enhanced detectability and robustness|X|X|✔️|
> |Supports embedding/extracting multi-bit messages|X| X|✔️|
> |Model-agnostic detectability|✔️|X|✔️|

---

### Official Review · Reviewer_FxLk · 2025-03-11

**Overall Recommendation:** 4

**Summary:**

This paper introduces BiMark, a comprehensive framework for watermarking large language models that achieves three critical objectives: text quality preservation, model-agnostic detection, and message embedding capacity. The core innovation is a novel probability distribution reweighting method with a multilayer architecture. The approach iteratively adjusts probability distributions across random vocabulary partitions using pseudorandom bits, keeping the modified probability consistent with the original probability in expectation to maintain text quality. The authors further develop methods for encoding and extracting messages within these patterns. Through theoretical analysis and experiments, the paper demonstrates BiMark’s ability to embed multi-bit information without degrading text quality, along with enhanced robustness to existing methods.

**Claims And Evidence:**

The claims made in the submission are well-supported by convincing evidence through both theoretical analysis and experimental results. Lemma 4.1, Theorem 4.2, and Appendices A.1 and A.2 provide theoretical guarantees for the unbiasedness of reweighting and the detectability of the watermark. Sections 4.3 and 4.4 describe the information embedding methods, and Figure 6 and Algorithm 1-2 in Appendix C detail the complete algorithmic process.  The experiments in Section 5 demonstrate the method's ability to carry multi-bit information and the impact of watermarks on text quality, and include ablation experiments to show the impact of multi-layer mechanisms on the detectability and robustness of watermarks.

**Essential References Not Discussed:**

None.

**Experimental Designs Or Analyses:**

The paper’s experimental design is sound, though the attack scenarios in robustness experiments could be more diverse.

**Methods And Evaluation Criteria:**

The proposed methods and evaluation criteria are appropriate for the watermarking problem addressed. The experiments comprehensively evaluate key aspects: message embedding capacity, text quality preservation, and watermark detectability.

**Other Comments Or Suggestions:**

The paper contains several typographical inconsistencies:

1) In Figure 3, the authors use the term “level” in the diagram but use “layer” throughout the rest of the paper.

2) In Figure 4 legends, the method name “KGW” appears, but “Soft Red List” is used in the paper.

**Other Strengths And Weaknesses:**

**Strengths**:

1.	   Introduces a novel vocabulary-based unbiased reweighting method using a coin-flip-like random variable to control probability redistribution direction, maintaining expected probabilities while creating detectable watermark patterns.

2.	   Proposes an innovative multilayer mechanism where multiple independent vocabulary partitions and unbiased reweighting allow each token to be influenced by multiple green lists, providing more observable evidence for watermark detection.

3.	   Creatively employs one-time-pad cryptography to encode messages into the watermarking control sequence, maintaining the unbiased property while carrying hidden information.

4.	    Provides both theoretical and experimental analysis of the unbiasedness and the detectability of the watermark.

**Weaknesses**:

1.	    Limited discussion of computational resources required for the multilayer unbiased reweighting method, particularly as layers increase.

2.	    Robustness experiments lack sufficient diversity of attack scenarios.

**Questions For Authors:**

1.	What is the computational impact of the multilayer architecture, particularly as the number of layers increases?
2.	How should scaling factors for unbiased reweighting be selected, and what factors does this selection depend on?

**Relation To Broader Scientific Literature:**

The paper properly situates its contributions within the existing literature:

1.	 [1] introduced using pseudorandom number generators with previous tokens as seeds for vocabulary bipartition during inference, adjusting token probabilities to inject watermarks, with detection via z-test of green token proportions.

2.	 [2] introduced unbiased probability distribution reweighting for preserving text quality, using cumulative probability thresholds for vocabulary partitioning, but requiring model access for detection.

3.	 [3] extended [2] with more refined probability reweighting but didn’t address message embedding.

4.	 [4] extended [1] to enable message embedding through encoding vocabulary partition.

BiMark naturally combines the red-green list partitioning from [1] with a novel unbiased reweighting method with a multilayer mechanism to achieve message-carrying, model-agnostic detection while preserving text quality.

[1] Kirchenbauer, John, et al. "A watermark for large language models." International Conference on Machine Learning. PMLR, 2023.

[2] Hu, Zhengmian, et al. "Unbiased Watermark for Large Language Models." The Twelfth International Conference on Learning Representations.

[3] Wu, Yihan, et al. "A Resilient and Accessible Distribution-Preserving Watermark for Large Language Models." International Conference on Machine Learning. PMLR, 2024.

[4] Yoo, KiYoon, Wonhyuk Ahn, and Nojun Kwak. "Advancing Beyond Identification: Multi-bit Watermark for Large Language Models." Proceedings of the 2024 Conference of the North American Chapter of the Association for Computational Linguistics: Human Language Technologies (Volume 1: Long Papers). 2024.

**Theoretical Claims:**

The theoretical is convincing.

---

> ### Author Rebuttal · Authors · 2025-04-01
>
> **Q1**: Computational cost of multi-layer reweighting
>
> **R1**: Please refer to our response to Reviewer jksy in *R1*.
>
> **Q2**: Robustness experiments
>
> **R2**: We conducted advanced paraphrasing attacks to our watermarked text.
> The detectability of BiMark under paraphrasing shows desirable robustness.
> Analysis reveals multi-layer watermarking provides fine-grained watermark evidence to keep the green ratio of watermarked text steadily deviating from 1/2 after paraphrasing.
> For more details, please refer to our response to Reviewer kPVR in *R3*.
>
> **Q3**: Terminology consistency
>
> **R3**: Thank you for pointing out the inconsistencies in our terminology. We will correct these issues and thoroughly revise the paper to avoid other potential issues.
>
> **Q4**: Scaling factor selection
>
> **R4**: For selecting scaling factors, we recommend adjusting based on the application scenario:
> - For better text quality: use smaller scaling factors with more layers.
> - For computational efficiency: use larger scaling factors with fewer layers.
>
> The underline insight behind these selection lies in entropy reduction effect of multi-layer reweighting. For details, please refer to our response to Reviewer jksy in *R2*.

---

### Decision · Program_Chairs · 2025-05-01

**Decision:**

Accept (poster)

**Comment:**

All three reviewers recommended to accept this paper in their reviews. After discussion, it was eventually agreed that this paper meets the ICML acceptance bar.
The authors are encouraged to address the comments of the reviewers to improve this work.
The author’s rebuttal has been carefully read and discussed. The author’s message has been carefully read and considered.